# The confounding effects of eye blinking on pupillometry, and their remedy

**Kyung Yoo, Jeongyeol Ahn, Sang-Hun Lee** *

Department of Brain and Cognitive Sciences, Seoul National University, Seoul, South Korea

* visionsl@snu.ac.kr

## Abstract

Pupillometry, thanks to its strong relationship with cognitive factors and recent advancements in measuring techniques, has become popular among cognitive or neural scientists as a tool for studying the physiological processes involved in mental or neural processes. Despite this growing popularity of pupillometry, the methodological understanding of pupillometry is limited, especially regarding potential factors that may threaten pupillary measurements' validity. Eye blinking can be a factor because it frequently occurs in a manner dependent on many cognitive components and induces a pulse-like pupillary change consisting of constriction and dilation with substantive magnitude and length. We set out to characterize the basic properties of this "blink-locked pupillary response (BPR)," including the shape and magnitude of BPR and their variability across subjects and blinks, as the first step of studying the confounding nature of eye blinking. Then, we demonstrated how the dependency of eye blinking on cognitive factors could confound, via BPR, the pupillary responses that are supposed to reflect the cognitive states of interest. By building a statistical model of how the confounding effects of eye blinking occur, we proposed a probabilistic-inference algorithm of de-confounding raw pupillary measurements and showed that the proposed algorithm selectively removed BPR and enhanced the statistical power of pupillometry experiments. Our findings call for attention to the presence and confounding nature of BPR in pupillometry. The algorithm we developed here can be used as an effective remedy for the confounding effects of BPR on pupillometry.

## Introduction

Even when ambient light is controlled, our pupil keeps dilating and constricting as mental operations or internal states of various sorts transpire in the course of carrying out diverse cognitive tasks [1–8]. This pupil-size dynamics is known to be tightly coupled with the activation of norepinephrine-containing neurons in the locus coeruleus (LC-NE system) and in other brain regions associated with the LC-NE system such as the colliculi and cingulate cortex [9–12]. Moreover, thanks to recent technical advancements, pupil size can be measured with high temporal resolution in non-invasive manners even when animals or humans are allowed to move their eyes and head relatively freely. For these reasons, pupil size became a popular physiological measure among cognitive scientists and neuroscientists, being considered a "peripheral window" into internal cognitive and neural processes [13,14].

**Data Availability Statement:** All data files are available from the OSF database (https://osf.io/uvs2d/).

**Funding:** This work was supported by the National Research Foundation of Korea Grants NRF-

2015M3C7A1031969, NRF-2018R1A4A1025891, and NRF-2017M3C7A1047860 All authors received the grant http://nrf.re.kr/ The funders had no role in study design, data collection and analysis, decision to publish, or preparation of the manuscript.

**Competing interests:** The authors have declared that no competing interests exist.

Despite the growing need and popularity of pupillometry, there is a lack of research on eye blinking as a confounding factor in pupillary experiments. Intriguingly, the pupil constricts and re-dilates within a few seconds after each event of eye blinking (Fig 1A, Supplementary S1 Video). This phenomenon, which will be referred to as "blink-locked pupillary response (BPR)" hereafter, is, in itself, not a discovery [15]. Although it is not entirely clear what causes BPR, the visual changes caused by the pupil's transient occlusion by the eyelid are considered its most likely source [16,17]. However, the presence of BPR seems not well known among researchers using pupillometry. They are not referred to in most pupillometry studies, including where the relationship between blinks and pupil size was investigated [18,19]. Even when BPR was mentioned, BPR was described as an "unexpected" phenomenon, and the data that were considered contaminated by BPR were simply discarded from further analysis [20].

There are reasons to believe that BPR should be considered a serious confounder that potentially threatens pupillometry experiments' internal validity or statistical power. First, humans blink spontaneously 3–15 times per minute [21–24]. Because each of such frequent blinks engenders BPR with substantial magnitude (constriction about 0.1–0.3 mm on average) and length (at least 2–3 s), a significant fraction of raw pupillometry data is expected to be dominated by BPR. Secondly, these seemingly "spontaneous" blinks do not occur randomly. Instead, the blink rate and patterns are highly dependent on subjects, task structures, and cognitive factors [25–28]. For example, blinks tend to be suppressed during task epochs important for task performance, such as when a target is presented, and concentrated around implicit breakpoints of a task, such as right after a response is made [19,29–31]. Furthermore, the blink rate depends on the activity of the dopamine (DA) system [32–34] or the executive functions that are known to be modulated by the system [35–38]. These studies suggest that BPR is likely to interact with cognitive factors of researchers' interest and the task or subject components of a given experimental design and that a substantial fraction of pupillary measurements is likely to be under the influence of such interactions.

Being motivated by these important implications of BPR on pupillary measurement, we set out to conduct a series of pupillometry experiments to understand how BPR confounds pupillary responses and figure out the proper way of handling BPR. Specifically, we proceeded as follows. First, by acquiring several BPR samples without asking subjects to perform any cognitive tasks, we learned about the general profile of BPR and how that profile varies across individual subjects and blinks. Second, by analyzing how the blink rate and pattern are associated with task structures and cognitive states, we learned how BPR interferes with the pupillary responses that are supposedly associated with the task components and cognitive variables of interest. Third, based on this learning, we built a statistical, generative model of how pupillary measurements are stochastically determined by the cognitive-state, spontaneous-state, and blink variables. Fourth, we developed a probabilistic-inference algorithm of correcting raw pupillary measurements for BPR with this generative model in hand. Lastly, we verified the effectiveness of the developed algorithm in selectively filtering out BPR and demonstrated that the algorithm could enhance the experiments' statistical power by applying it to the data in the cognitive pupillometry experiments. Our studies suggest that BPR should not be treated as a mere nuisance or trivial phenomenon that can be averaged or deleted away, but rather as a serious confounder, something that potentially leads to misinterpretation or reduced statistical power of experimental results. Our correction algorithm illustrates one proper way of handling BPR for a wide range of cognitive experiments using pupillometry.

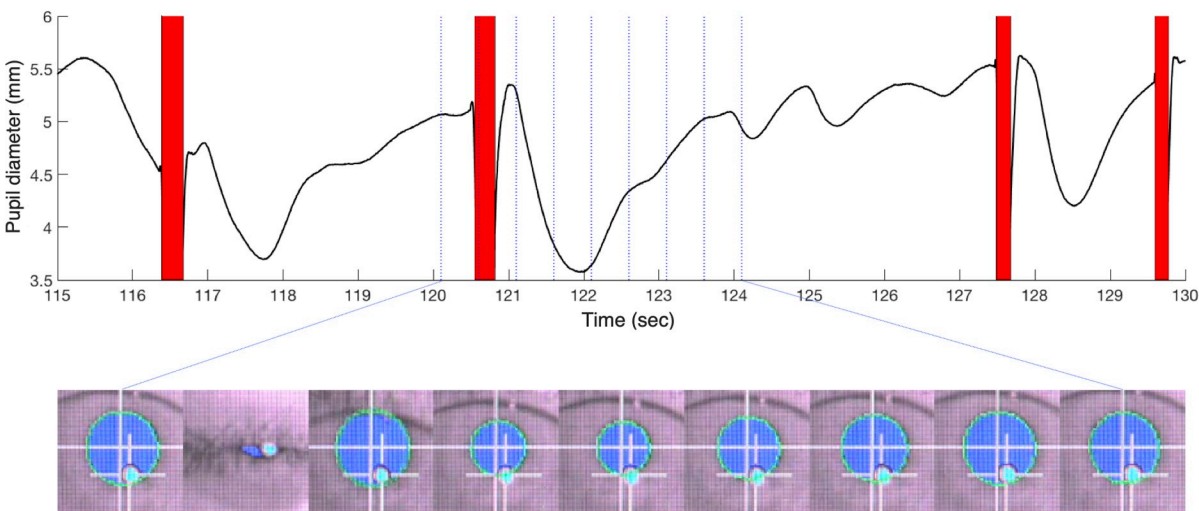

**Fig 1. Example of blink-locked pupillary response (BPR).** Raw pupil diameter time course example (top panel), and snapshots of recorded eye video (bottom panel). At the top panel, red bars indicate blinks, and blue vertical lines indicate the time of each eye image snapshot. At the bottom panel, the blue circle is a pupil area defined by the eye tracker, and the cyan circle is a corneal reflection. The snapshots are 4-s long and 2Hz frame rate. It clearly shows blink-locked pupillary response (BPR), that pupil constricts and re-dilates transiently after blinking. See the video (S1 Video) for vivid BPR examples.

## Results

### The shape of blink-locked pupillary responses and its variability across luminance levels, subjects, and trials

The first goal of the current work is to describe the generic shape of BPR and to assess whether and, if so, how the shape of BPR varies depending on experimental factors, including background-luminance levels, subjects, and trials. To keep the other factors from influencing BPR, we measured the pupil size while subjects kept fixating at the center of the screen without performing any cognitive tasks at all (Fig 2A). To avoid any unwanted changes in cognitive state due to intentional suppression of natural blinks [39], we did not give subjects any instructions about eye blinking (e.g., "when they are, or are not, allowed to blink"; "how often they can blink"), except "blink naturally." To define the shape of BPRs induced by sufficiently isolated single blink events, we only used the BPR samples associated with the blinks that were apart more than 3 s from their nearest blink (see Materials and Methods for details).

In line with previous work [15], the pupil size initially constricted right after an eye blink and returned to the baseline size (Fig 2B). When averaged across the luminance conditions, subjects, and trials, the BPR reached its peak constriction (−0.2 mm) at about 0.9 s after an eye blink and returned to its baseline size at about 3.0 s after an eye blink (Fig 2C). Given that the difference in pupil diameter by a manipulated variable in general pupillometry usually ranges from 0.05 to 0.5mm [40], pupil diameter change by BPR was considerable.

We found that the general shape of BPR, i.e., an initial fast constriction followed by a slow dilation back to the baseline size, remained unchanged, but its peak amplitude and time-to-the-peak substantially varied across different luminance levels, individual subjects, and individual blinks. The peak amplitude increased as the background luminance increased up to the second-highest level (5.35 cd m$^{-2}$) but somewhat decreased under the highest level (97.60 cd m$^{-2}$; Fig 2D). This result is consistent with the view that BPR reflects the pupillary response to transient changes in retinal illuminance [15,41]. The slight decrease under the highest level of

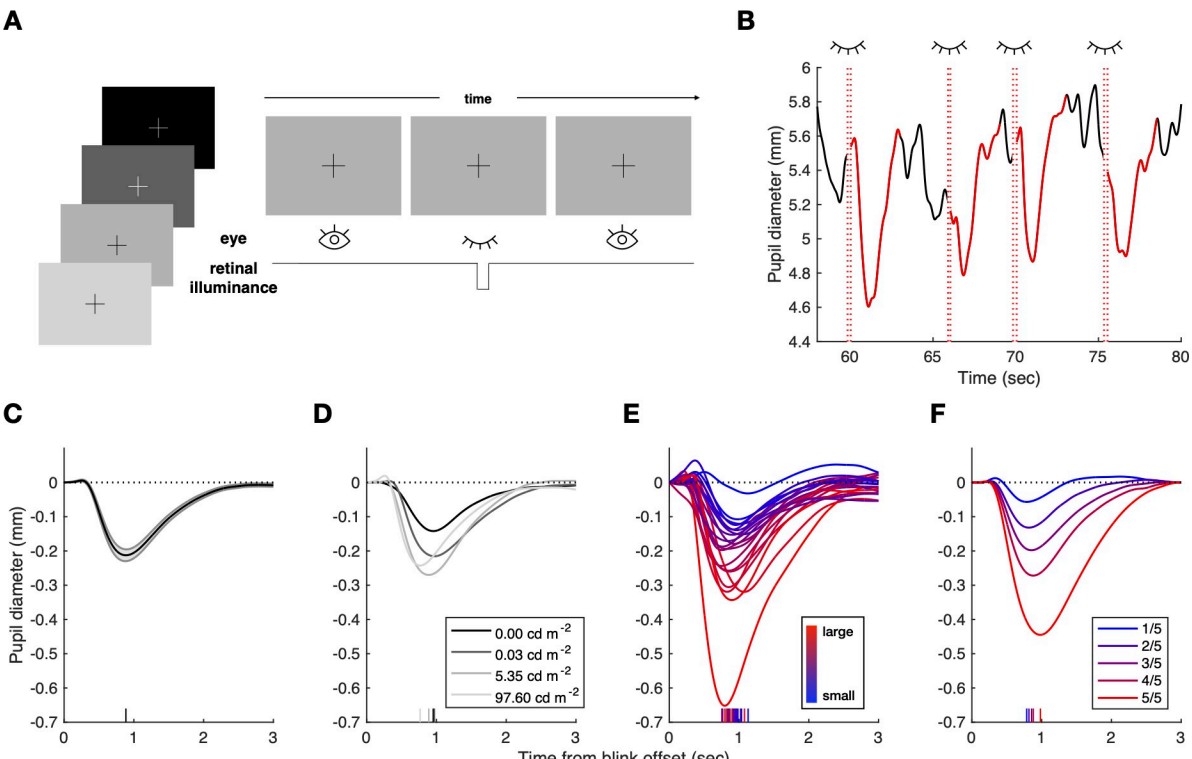

**Fig 2. Experimental design and examination of factors affecting blink-locked pupillary responses.** (**A**) The fixation task was designed to measure BPR without the influences from other sources, such as saccade or other task-evoked components, as much as possible. In the experiment, four background-luminance levels were used. There was no stimulus, and luminance remained constant. Therefore, retinal illuminance decreases only when subjects blink. (**B**) Pupil and blink time series in an example run. The red dotted line indicates the blink onset or offset. Note that the pupil constricts and re-dilates transiently after blinks. (**C–E**) BPR time courses varying across several factors. Small vertical lines indicate the time-to-peak of each time course. (**C**) Grand average of BPR across subjects and background luminance. Line and shade indicate the mean and standard errors of the mean (SEM) across subjects and luminance, respectively. (**D**) BPR average across subjects within the background-luminance levels. BPR depended on the background luminance. The legend indicates the corneal flux density of the visual field ($cd\ m^{-2}\ deg^2$). (**E**) BPR average across background-luminance levels within subjects. BPR shape and size were idiosyncratic across subjects. The lines were colored more reddish as their negative peak becomes larger. (**F**) BPR profiles are sorted into five quantiles by their peak amplitudes. Note that BPR amplitude is different blink-by-blink even when the effect of subject and luminance were ruled out. Also, note that the characteristic short-pipe shape of BPR maintains under any conditions.

luminance appears to reflect the fact that the baseline pupil size (mean ± SD across subjects, 3.13 ± 0.51 mm) was likely to be close to the minimum pupil size [42] so that BPR constriction was bounded.

Both the peak amplitude and the time-to-the-peak of BPR also varied substantially across individual subjects (Fig 2E). The shape of BPR appears idiosyncratic to each individual, as supported by the high degrees of split-half (even versus odd runs) correlation in both peak amplitude (Pearson correlation r > 0.9 under any backgrounds) and in time-to-the-peak (r > 0.7 under any backgrounds) (see S1 Fig for details).

Lastly, the blink-by-blink variability of peak amplitude was also substantial. To examine the blink-by-blink variability independent of luminance level and idiosyncrasy across subjects, BPRs were sorted into five quantiles by their peak amplitudes under balanced effects of luminance level and subject (see Materials and Methods for details). The peak amplitude of the highest quantile was 7.85 times greater than the peak amplitude of the lowest quantile (Fig 2F). It means that the peak amplitude of each blink varies considerably within the same background-luminance level and subject.

To summarize, single blinks instigate a three-second-long BPR in which the pupil quickly constricts and then slowly dilates back to its baseline size. The shape of BPR, especially the peak amplitude, varies substantially across luminance levels, subjects, and blinks.

## BPR, not just a nuisance but a confounder

We argue that BPR not only makes pupil-size measurements noisy but also can confound them in various manners if eye blinks, the trigger of BPR, do not occur randomly, but their frequency is systematically dependent on task phases or experimental variables. In such cases, BPR can threaten the internal validity of pupillometry experiments. Previous work suggests such dependencies of eye blinks. In humans, eye blinking does not occur just reflexively such as when the eyes need to be lubricated [43]. Instead, the rate of spontaneous blinks is closely associated with cognitive factors [22]. For instance, eye blinking tends to be held during the task epochs important for information processing (e.g., target presentation) and then vigorously occurs during the intervals between those "important" epochs, which are often dubbed "implicit breakpoints" [19,29–31]. The rate of spontaneous blink also depends on the activity of the dopamine system in the brain [32–34] or on the executive functions and reward processes, which are modulated by the dopamine system [35–38].

To illustrate how BPR can confound the pupillary signal of interest, we measured pupil size while subjects performed two cognitive tasks, "auditory oddball detection" and "delayed orientation estimation" tasks. These two tasks were chosen because the cognitive processes underlying the tasks, i.e., 'surprise by oddity' and 'working memory load', are relatively well established, and those processes are known to be tightly associated with pupillary responses [5,44–47].

In the auditory oddball task, in which subjects had to discriminate the sounds of two tones every 2 s (Fig 3A; see Materials and Methods for details), the length of the trial (2 s) is shorter than the length of BPR (3 s). This implies that the impact of single blinks on pupillary responses is not confined to current trials but extended to subsequent trials, which complicate the time course of pupillary responses such that the application of conventional analyses to raw pupil data leads to serious misinterpretation. To illustrate such confounding effects, we simulated synthetic pupillary responses by assuming that each blink generates a canonical BPR, the average of isolated BPR samples collected under the second-highest level of the fixation task (Fig 2D), where retinal illuminance was most similar, and there is no change in the pupillary responses other than BPRs. Specifically, we simulated the time courses of pupillary responses in many pairs of two consecutive trials. In line with previous studies that showed suppression of blink rate during target stimulus presentation (or other cognitively important moments) and concentration of blink rate during implicit breakpoints [19,29–31], the blink rate was low during the stimulus presentation and increased rapidly as well, and slowly dwindled afterward (Fig 3B, top). When averaged over many simulated pairs of trials, the occurrences of blinks with that distribution resulted in the pupillary responses that (negatively) peaked at around the boundary between the first and second trials and returned to the baseline level at around the end of the second trial (Fig 3B, bottom). These simulation results imply that BPR will make pupillary responses deviate from the true pupillary signal of interest (the black dotted line in Fig 3B, bottom). Such deviations due to BPR, when combined with the procedure of setting the baseline of single-trial pupillary responses to the pupil diameter at the trial onset, which is conventionally practiced, will result in large spurious biases, decreasing and increasing time courses of pupillary responses in the blinking (first) and post-blinking (second) trials, respectively.

In the delayed orientation estimation task, we asked subjects to estimate the orientation of a post-cued target after a short delay while varying the number of bars to be remembered (Fig

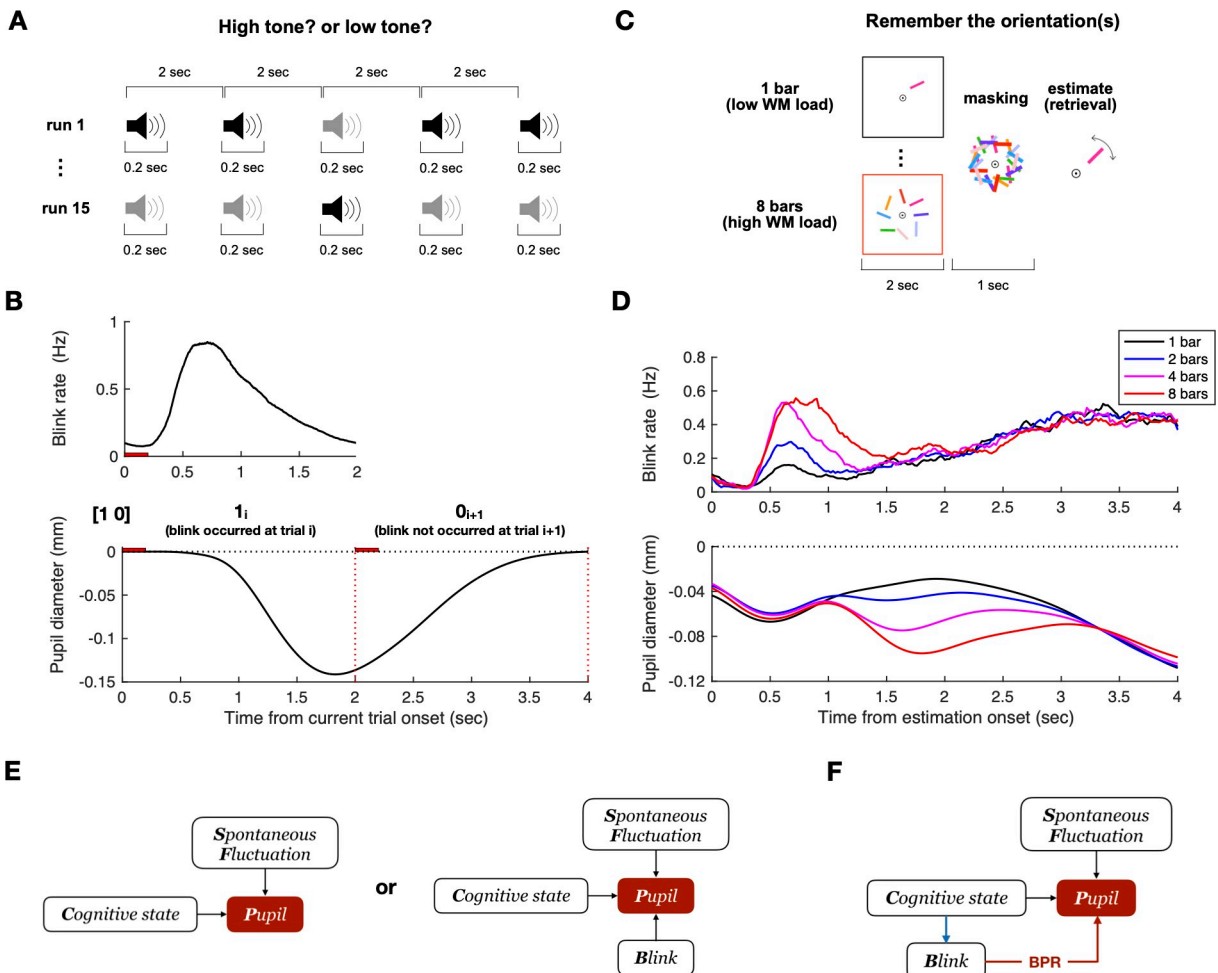

**Fig 3. BPR is predicted to bias pupil size because blink rate and pattern vary across time, working-memory load, and subjects.** (**A**) Auditory oddball task structure. 0.2 s long high and low-tone beep sound presented every 2 s. Subjects are instructed to press one of the two buttons corresponding to low and high tones as soon and accurately as possible. The high-tone to low-tone presentation ratio is 2:8, 5:5, or 8:2, and the ratio is randomized across runs. (**B**) Blink rate mean and predicted BPR confound pattern in the auditory oddball task. The red horizontal bar and vertical lines indicate the presentation time of the auditory stimulation and the end of each trial, respectively. Note that BPR confound lasts till the subsequent trial. The pupil size decreased in the blink-occurred trial and recovered in the ensuing trial. (**C**) Delayed orientation estimation task structure. Bar (s) are presented 2 s long. It has four levels of difficulties to manipulate the degree of working-memory load (1,2,4 and 8 bars). Following a mask, subjects had to retrieve the memory to estimate the orientation of a post-cued bar. (**D**) Blink rate mean and predicted BPR confound pattern in the delayed orientation estimation task. The time 0 is the onset of the estimation epoch. The color indicates the number of bars to be memorized (N = 1,2,4 and 8 are denoted as black, blue, magenta, and red lines, respectively). Note that blink rate increased as the memory load increased, and thereby BPR is predicted to be biased to high memory load conditions. (**E**) Implicit models of pupillometry. They do not consider the effect of blink on pupil size at all (left), or treat the effect as a nuisance (right). (**F**) Realistic model of pupillometry. It considers blinks as confounders because blinks provide a backdoor from the cognitive-state variable to the pupil-size variable.

3C; see Materials and Methods for details). Suppose the independent variable of interest is the number of bars to be remembered (i.e., visual working-memory loads). Intriguingly, we found that the actual rate of eye blinking systematically varied depending on the number of to-be-remembered bars during the estimation epoch: the blink rate tended to increase with an increasing number of bars (Fig 3D, top). When the synthetic pupillary responses were simulated as were done for the auditory oddball task data, the differences in the blink rate between the conditions resulted in the mirroring, spurious differences in pupillary responses: the negative peak amplitude of pupil size increased with an increasing number of bars (Fig 3D,

bottom). These simulation results imply that BPR can confound the pupillary responses of interest—i.e., those associated with "visual working-memory load"—with the blink rate because BPR differentially biases the pupil size depending on the blink rate that happens to be associated with the independent variable of interest.

Put together, the simulation results show that BPR can readily confound the pupil responses of interest, i.e., those associated with a cognitive state of interest, whenever the blink rate changes depending on the cognitive state of interest. It is important to be aware of such confounded-by-BPR pupillary responses and treat them properly because they will be directly translated into the "signals of interest" by researchers who are unaware of the presence of BPR per se (Fig 3E, left), which severely threatens the internal validity of findings leading to either type I (false positive) or type II (false negative) errors in statistical inference. If researchers are aware of BPR but do not fully recognize the confounding nature of BPR (Fig 3E, right), they might simply treat BPR as a noisy nuisance by getting rid of the trials that are affected with blinks selectively. Unfortunately, this "selective screening" approach cannot "de-confound" the pupillary responses for BPR but instead may invite other confounders of a complicated nature because such a selection itself is confounded with eye blinks, which confounds a cognitive state of interest.

## Model-based correction of pupillary responses for BPR

From the fixation task, we have learned about the shape and variability of BPR (Fig 2). From the changes in the blink rate during the auditory oddball task and delayed orientation estimation tasks, we have also learned about the confounding nature of BPR (Fig 3). Based on this discovery, we built a generative model of pupillary responses. According to this model (Fig 4A, left), a pupillary measurement is a random variable that has three random variables as parents, namely, the "cognitive-state," "spontaneous-state," and "blink" variables. Based on what was discovered from the fixation task, we approximated the causal function from the blink variable to the pupil size (i.e., BPR) with an inverted gamma probability density function, the parameters of which are unique to a given individual and the amplitude of which varies across blinks. Importantly, we incorporated the confounding nature of BPR into the generative model by positing that the cognitive state has another indirect causal route to the pupil size through the blink variable.

Given this generative model of pupil-size measurements, the normative way of de-confounding pupil measurements must block the backdoor route from the cognitive state to pupil measurement via the blink variable. Blocking this back door means inferring "counterfactually" what pupil measurements would be like if blinks did not occur. This counterfactual inference can be achieved only when we can estimate the subject-specific shape and the blink-specific amplitude of BPR and then take out that specific BPR selectively from the raw measurements of the pupil size (Fig 4A, right). Such selective filtering will leave the causal influences from both the cognitive state and the spontaneous state intact in pupil measurements.

We stress that our intended purpose of correction is not to infer the cognitive state per se but to get rid of the influence of blink events selectively on pupil measurements so that the corrected pupil measurements still reflect the influences both from the cognitive state and from the spontaneous state. We intended so not simply because it is difficult to distinguish the influence from the spontaneous state from the cognitive state but rather because the spontaneous state itself can be of interest to some researchers (e.g., those who want to know the impact of the spontaneous state on pupil measurements).

The critical part of our correction method is to accurately estimate the subject-specific shape parameters and the blink-specific amplitude parameter of BPR ($BPR_{js}$ in Fig 4A). By incorporating what we learned about BPR from the fixation task into the generative model as

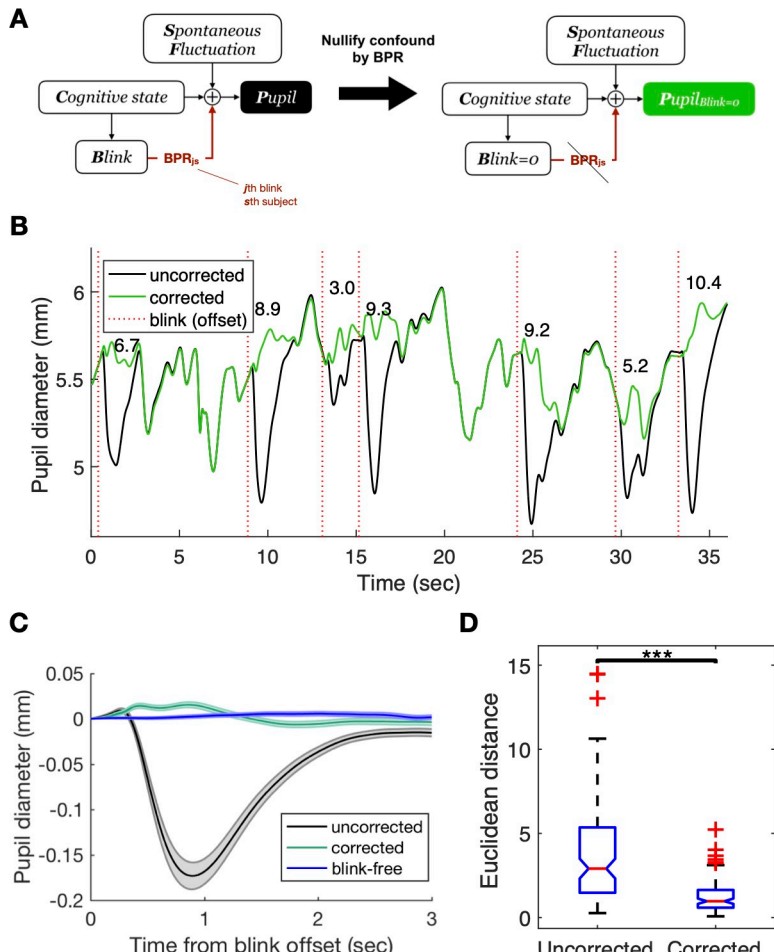

**Fig 4. Generative model-based decomposition of pupil-size measurements and example correction result.** (**A**) According to the realistic model, observed pupil size is an amalgam of response to cognitive state, BPR, and spontaneous fluctuation from other cognitive-independent sources. The goal is to decompose the pupil size as the sum of the components with a generative model and eliminate BPR specifically based on the estimated BPR component, making the data free from BPR confound. The algorithm estimates the BPR shape for each subject and BPR amplitude for each blink (e.g., $s$th subject's $j$th blink is estimated here) (see Materials and Methods for the detailed description of the algorithm). (**B**) An example of BPR correction results. Note that the correction algorithm estimates the BPR amplitude for each blink (denoted as numbers above time course) based on the generative model. (**C**) Comparison between blink-locked time course from pre-BPR correction (black), post-BPR correction (green), and blink-free period (blue). Note that the characteristic short-pipe shape of BPR before correction is flattened and resembles a blink-free period after being corrected for BPR. Line and shade indicate the mean and standard errors of the mean (SEM), respectively. (**D**) Euclidean distance between the mean blink-locked time course and the mean blink-free time course in each subject. The distance decreases significantly after BPR correction ($p < 0.001$; ***).

prior knowledge, we developed an algorithm inferring subject-specific BPR profile and blink-by-blink BPR amplitude by combining that prior knowledge and the likelihood function acquired from the observed pupil measurements (see Materials and Methods for the detailed description of the algorithm). Given the substantial blink-to-blink variability of BPR amplitudes (Fig 2F), we stress the importance of estimating the amplitude of BPR on a blink-to-blink basis. In this regard, our algorithm is distinguished from and goes beyond the previously proposed correction method that assumed the constant amplitude of BPR [17].

As an initial step of validating our correction algorithm, we applied it to the data acquired from the fixation task (Fig 4B). One important merit of this data set is the fact that an effective

ground-truth time course of pupillary responses, which are free from BPR, can be approximated by the average of the sufficiently long periods during which blinks did not occur (see Materials and Methods for the detailed description of blink-free time courses). It is important to compare the blink-affected time course of pupillary responses against this ground-truth time course before and after, respectively, the correction for BPR because it allows us to verify whether our correction algorithm selectively removes the pupillary responses associated with BPR but not those associated with other components, including cognitive and spontaneous fluctuations. In this regard, note that the blink-free, ground-truth time course of pupillary responses might not be necessarily flat because there could be many uncontrolled factors that might affect the pupillary responses (e.g., gradual decrement or increment in pupil size over time due to fatigue or arousal). In such cases, the simple comparison of the pupillary responses before and after the application of BPR correction does not provide sufficient information about whether the algorithm successfully corrected the pupillary responses only for BPR. If the algorithm succeeds in selectively removing BPR from the raw pupil-size measurements, the time course of the corrected measurements must be close to the ground-truth time course. This prediction was confirmed: the original average profile of the pupillary responses with blinks showed the typical shape and amplitude of BPR before correction (gray line in Fig 4C) but became close to the average profile of the pupillary responses without blinks after the correction (green and blue lines in Fig 4C). Statistical tests showed that the correction significantly reduced the distance between the profile of the pupillary responses with blinks and that without blinks (Wilcoxon signed-rank test $p < 0.001$) (Fig 4D).

## Correcting the auditory oddball data for BPR

As the second step of verifying the effectiveness of the correction algorithm, we applied it to the data acquired in the auditory oddball task. Unlike the fixation task, the auditory oddball task was designed to test a particular prediction of a specific cognitive hypothesis. For example, in line with previous studies [44,45], suppose an experimenter predicts that the pupil size will increase as the oddity increases according to the hypothesis that the oddity is associated with the degree of a surprise based on the strong relationship between pupil size and degree of surprise. To describe in terms of the generative model (Fig 3F), this means that the "cognitive-state" variable, i.e., "oddity," is manipulated by the experimenter and expected to influence the pupillary measurement variable in the auditory oddball task. Thus, unlike the fixation task in which no specific cognitive state was manipulated, the oddball task offers an opportunity to further verify the correction algorithm by testing whether our correction algorithm can selectively remove BPR in the presence of the influence of a cognitive state on pupillary measurements.

Before testing the algorithm's effectiveness, we first validated the generative model by testing how successfully it can predict the time course of raw pupillary responses in the presence of both cognitive and BPR influences on pupillary measurements. To do so, we simulated the time series of pupillary responses (as depicted by the black line in Fig 5A) by positing that a beep sound and an ensuing manual motor response together constantly generate an increasing and decreasing profile of pupillary responses (as depicted by the blue line in Fig 5A) and, in parallel, that an event of eye blinking also constantly generates the canonical BPR (as depicted by the red line in Fig 5A). The procedure for this simulation was identical to that for the one depicted in Fig 3B, except that the constant pupillary response associated with the sound and the manual motor response is linearly added (see Materials and Methods for details). As was done previously, to illustrate the varying impacts of BPR on the data acquired in the auditory oddball task, we contrasted the averaged, 2-second-long profile of simulated pupillary

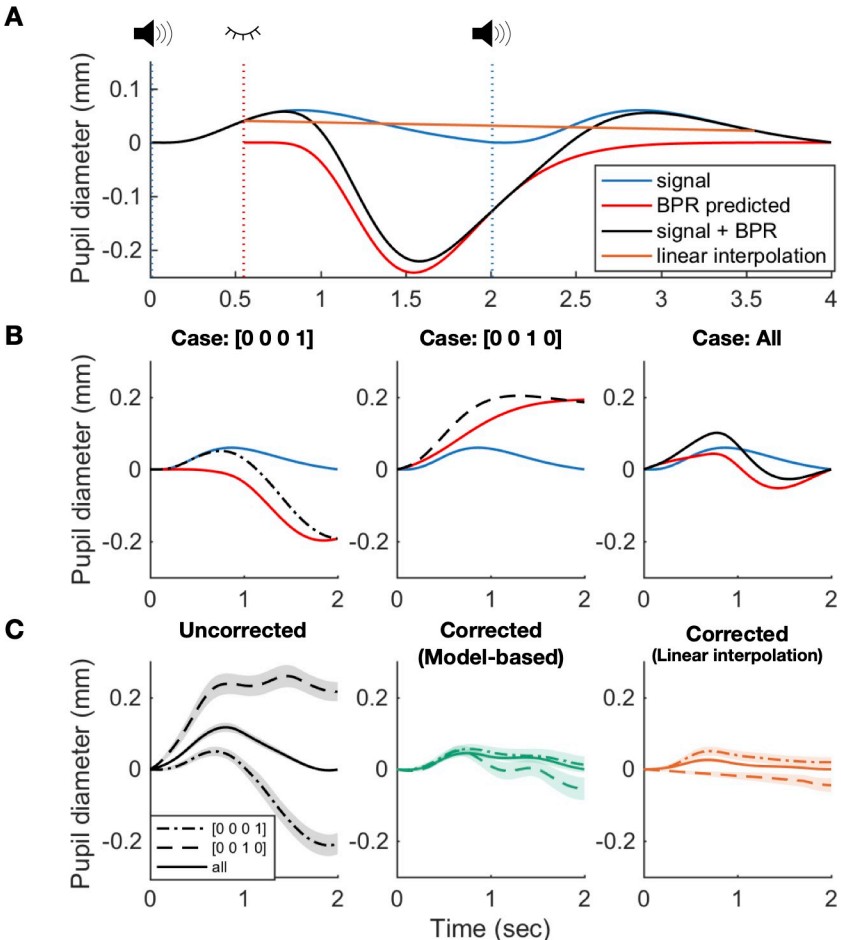

**Fig 5. Temporal bias in blink patterns induces spurious distortion of pupil time course in the auditory oddball task.** (**A**) Schematic example addressing how BPR confounds pupillometry in the auditory oddball task. The pupil dilates by an auditory tone, which is the signal experimenters are interested in (blue). However, subjects spontaneously blink during the experiment, and it evokes BPR (red). Subjects tend to frequently blink about 500 ms after the stimulus (shown in Fig 3B). Therefore, BPR would distort the pupil response to the stimulus by decreasing it during the trial in which blink occurred and by increasing it during the subsequent trial. Linear interpolation of 3 s after a blink (orange) slightly reduced the pupil response in this case. (**B**) Simulation of pupil time course shape by BPR. BPR would induce significant and spurious distortion in shape. To identify the spurious distortion pattern by BPR clearly, two cases were selected. The cases where a blink occurred at the current trial (left panel) or right before the trial (middle panel) are shown (see Materials and Methods for details). The observed pupil time course (black) would be the sum of the pupil response to cognitive state (blue) and BPR (red). Note that it has increased and decreased by BPR (red) as shown in A. (right panel) In all trials, the increasing and decreasing effects are combined and happened such that predicted mean profile of pupil-size measurements (blue) so somewhat exaggerated the peak of the pupil dilation. (**C**) The spurious distortion in shape was identified as anticipated and fixed by the correction method. (Left panel) Pre-correction pupil time courses (black). When a blink occurred, the spurious decrease in the blink-occurred trials (dash-dotted line), the spurious increase in the post-blink trials (dashed line), and the spurious increase for the first 1 s and the following decrease in all trials (solid line) were observed as anticipated by the generative model. (Middle panel) Most of these spurious distortions were removed or attenuated after applying the model-based BPR correction method (green). (Right panel) The spurious distortions were over-corrected after applying the 3-s long linear interpolation correction method such that the all-trial average profiles of pupil-size measurements was substantially reduced (orange). Line and shade indicate the mean and standard errors of the mean (SEM) across subjects, respectively.

responses for two cases: (i) in which blinking occurred on the current trial but neither previous nor following trials (left panel in Fig 5B); (ii) in which blinking occurred on the immediately preceding trial but not on the current trial nor 2, 3 back trials (middle panel in Fig 5B). Additionally, to illustrate the overall impact of BPR, we also plotted the grand average of simulated

pupillary responses (right panel in Fig 5B). For the first case, the generative model predicts that the profile of raw pupillary responses initially follows that of the sound-locked responses (signal) and then rapidly drops due to the constriction caused by BPR. By contrast, for the second case, the profile of raw pupillary responses is predicted to increase more rapidly than that of the sound-locked reactions and continues to increase due to the dilation caused by BPR, while that of the sound-locked responses decreases. Note that this exaggerated profile is the spurious outcome of the practice of adapting the baseline to trial onset (as indicated by the cross point between the dotted vertical blue line and black line in Fig 5A) in conjunction with the "return-to-baseline" dynamics of BPR that was triggered by the blink in the preceding trial (red solid line in Fig 5A). For the third case, the generative model predicts that BPR makes even the all-trial averaged profile of the raw pupillary responses substantially deviate from that of the sound-locked responses. The averaged profile of the raw responses is predicted to be initially greater and then smaller than that of the sound-locked responses. These predictions well matched the observed profiles for all three cases (as indicated by the sound correspondences between the simulated black lines in Fig 5B and the observed ones in the left panel of Fig 5C).

Having confirmed the predictions of the generative model upon which our correction algorithm is built, we turn to check whether our model-based counterfactual-inference method selectively filters out the biases due to BPR in the raw pupillary responses. If so, the corrected profiles of pupillary responses in the three cases must become much close to one another. The close matches between the observed profiles of the three cases (middle panel of Fig 5C) indicate that the biases due to BPR were successfully corrected. We stress that this selective removal of BPR cannot be accomplished simply by deleting the raw pupil-size data under the influence of BPR and linearly interpolating those deleted data (as indicated by the orange line in Fig 5A). The linear interpolation method appeared to reduce the biases due to BPR to some degrees but invited the systematic errors opposite to those observed in the uncorrected data: the amplitude of the interpolated profile was increased and decreased for the first and second cases, respectively, compared to the third case (Fig 5C, right). This pattern can readily be expected from the spurious effects due to baseline setting (the orange line in Fig 5A).

As another way of verifying the effectiveness of the proposed algorithm, we sorted trials into those that were affected by blinks ("blink-affected" trials) and those that were free from blinks ("blink-free" trials), and compared the two types of trials in pupil-size time course before and after the correction algorithm was applied, as we have done in the fixation experiment. The rationale and prediction of this comparison analysis are the same as those stated in the fixation experiment: If the algorithm succeeds in selectively removing BPR from the raw pupil-size measurements, the time course of the corrected measurements must be close to the time course of blink-free, which is a proxy of ground-truth signal, after BPR correction. Considering that BPR lasts for about 3 s, the blink-free trials were defined as the trials whereby blinks occurred neither in the 2-back trial, in the 1-back trial, nor in the current trial while the blink-affected trials were all the rest of the trials (see Methods and Materials for details). Using the time course of pupil-size changes in the blink-free trials as effective ground truth of pupillary responses to the cognitive event of interest, we examined: (1) how much the time course of pupil-size measurements in the blink-affected trials deviates from that in the blink-free trials; (2) whether, and how effectively if so, the correction algorithm reduces such deviations. We found that the time course of pupil size in the blink-free trials (gray lines in Fig 6A) exhibits double peaks. The first and second peaks can be interpreted to reflect the sound stimulus and the following manual response. This interpretation is consistent with previous reports where similar double-peak pupillary responses were reliably observed in the auditory oddball tasks in which subjects made a manual response on each and every trial like in our study [44,48]. Before the correction, the time course of pupil size was significantly exaggerated in the

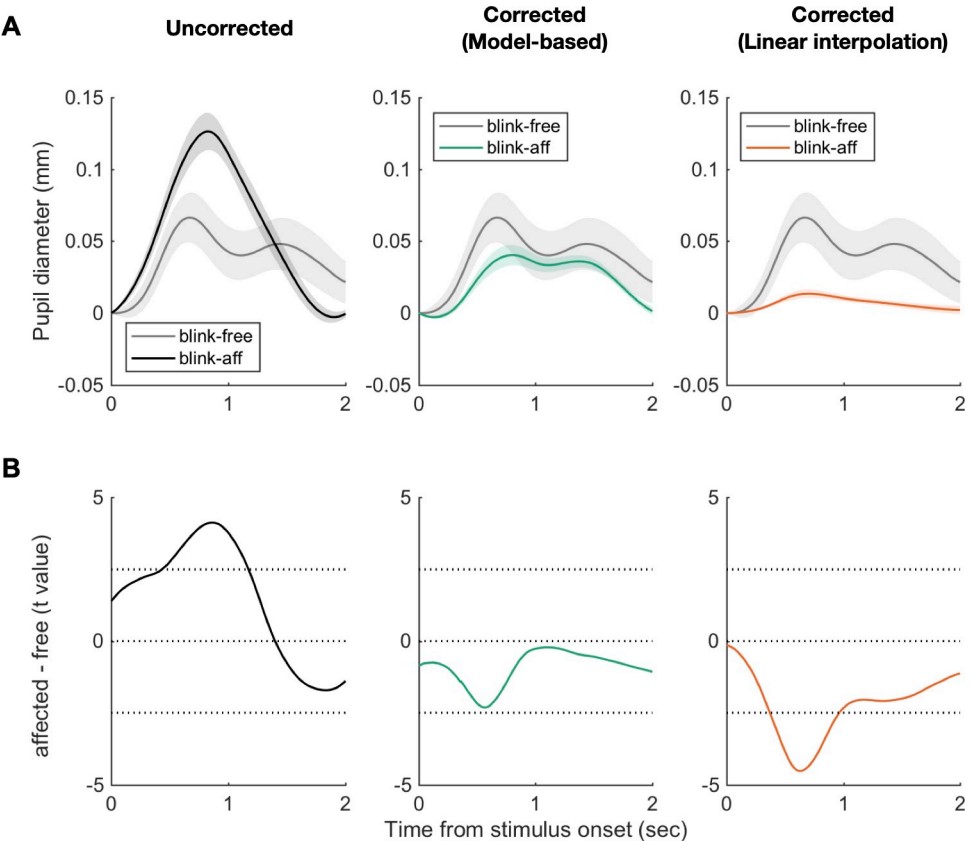

**Fig 6. Comparison of the blink-affected trials and the blink-free trials before and after correction.** (**A**) The time courses of pupil size before (left) and after correction (middle and right). The gray lines represent the mean time course of pupil-size measurements in the blink-free trials. The black line is the mean time course of pupil size in the blink-affected trials before correction. The green and orange lines represent the mean time courses of pupil size after correction with the model-based method and the linear interpolation method. Shades represent standard errors of the mean (SEM) across subjects. (**B**) The time courses of t statistics ("blink-affected"–"blink-free," df = 23) of the difference between the blink-affected and the blink-free trials. Horizontal dashed lines indicate the values of t statistics where the significance level is 0.01, and $t = 0$.

blink-affected trials, particularly around the peak (left panels in Fig 6A and 6B). Some might think that this exaggerated increase is weird because BPR appears to 'increase' the pupil size. However, this increase can be anticipated because BPR lasts longer than a single trial, and the time course of pupil size is baselined. In the trials in which blinks occurred in the current trial, the effect of BPR appears as a dip. The effect usually affects only the second half part (1–2 s) of the trial (see 0–2 sec of the bottom panel of Fig 3B) since blinks are concentrated at 0.5–1 s and (as described in the top panel of Fig 3B) and BPR has some delay (0.3 s) before a substantial decrease. In the trials in which blinks occurred in the 1-back trial, the BPR from the 1-back trial typically reaches its negative peak around the onset of the current trial, so the baseline is spuriously decreased. As the pupil diameter at stimulus onset is set as the baseline, BPR in the recovery (dilation) phase appears as a spurious increase of the first half part (0–1 s) of the trial (see 2–4 sec of the bottom panel of Fig 3B). In the trials in which blinks occurred in the 2-back trial, the BPR effect is similar to that in the 1-back trial case, but with a smaller effect size since BPR from the 2-back trial substantially recovered prior to the current trial. The combination of the decrease in the last half part (from the trials in which blinks occurred in the current trial) and the increase in the first half part (from the trials in which blinks occurred in the

1-back and 2-back trial) eventually made spuriously exaggerated dilation of the baselined time course in this task. We stress that these spurious patterns were exactly those that were predicted by the simulation based on our generative model (Fig 5A and 5B), and the observed patterns (Fig 5C) confirmed this anticipation. The thick black line in Fig 6A, which is the same as the thick black line in the left panel of Fig 5C, is the observed time course of the "blink-affected" trials, which is quite similar to that predicted by the model simulation (the right panel of Fig 5B) and the mean of the spurious patterns (dashed and dash-dotted lines in the left panel of Fig 5C). This deviation became substantially reduced after being corrected by the model-based method (middle panels in Fig 6A and 6B). Specifically, the corrected time course in the blink-affected trials was similar to the time course in the blink-free trials in shape, exhibiting double peaks. However, in amplitude, the corrected time course in the blink-affected trials was slightly smaller than that in the blink-free trials. This slight deviation may be taken as a limitation of the correction method. However, alternatively, it may reflect the genuine difference between the blink-affected trials and the blink-free trials, which is a matter of empirical investigation. By contrast, when the interpolation-based algorithm was applied, the corrected time course in the blink-affected trials was significantly below the time course in the blink-free trials (right panels in Fig 6A and 6B). As demonstrated in Fig 5, this understated pattern can readily be explained by the limitation of the interpolation approach, which not only removes BPR but also throws out the genuine pupillary responses associated with the cognitive factors of interest (i.e., the sound and the manual response to it).

Lastly, we evaluated the utility of the correction algorithm by assessing the extent to which it contributes to the statistical power of revealing the differences in pupillary responses between the experimental conditions of interest, i.e., oddity levels in the auditory oddball detection task. This evaluation is important and cannot be guaranteed on the basis of the above test, which only demonstrated the algorithm's effectiveness in removing the mean bias due to BPR in specific cases. The statistical power, in principle, can be improved or deteriorated not just due to the mean difference between conditions but also due to the trial-to-trial variability. In this regard, it should be reminded that the key feature of our correction algorithm is to estimate the shape and amplitude of BPR on the subject-to-subject and blink-to-blink bases, respectively. Thus, if this estimation is successfully carried out, the statistical power must be improved. As an initial step of testing such improvement in statistical power, we examined how much the model-based method enhanced the statistical differences between the levels of the independent variable, i.e., the oddity of sound frequency, over the trial-locked time course of pupil size. As anticipated, the pupil dilation increased as a function of oddity even before correction (left panels in Fig 7A). However, after being corrected for BPR with the model-based method, the differences between the oddity levels substantially increased, especially in the later part of the time course (middle panels in Fig 7A). By contrast, the interpolation method substantially decreased the statistical differences between the oddity levels (right panels in Fig 7A). Next, we assessed the contributions of the correction algorithm to the changes in the mean difference and standard errors of the mean (SEM) across trials, respectively. We found that the correction algorithm did not affect the mean difference (green symbols in Fig 7B) but robustly decreased the SEM (green symbols in Fig 7C). As a result, the correction algorithm significantly improved the statistical power when the power was assessed both by the area-under-the-curve analysis (green symbols in Fig 7D; see Materials and Methods for details) and by the simulation analysis in which the minimum number of trials required to reject the null hypothesis was calculated (green line in Fig 7E; see Materials and Methods for details). In contrast with our correction algorithm, the interpolation method ended up decreasing the statistical power (orange symbols in Fig 7D; orange line in Fig 7E) by substantially reducing the mean differences (orange symbols in Fig 7B).

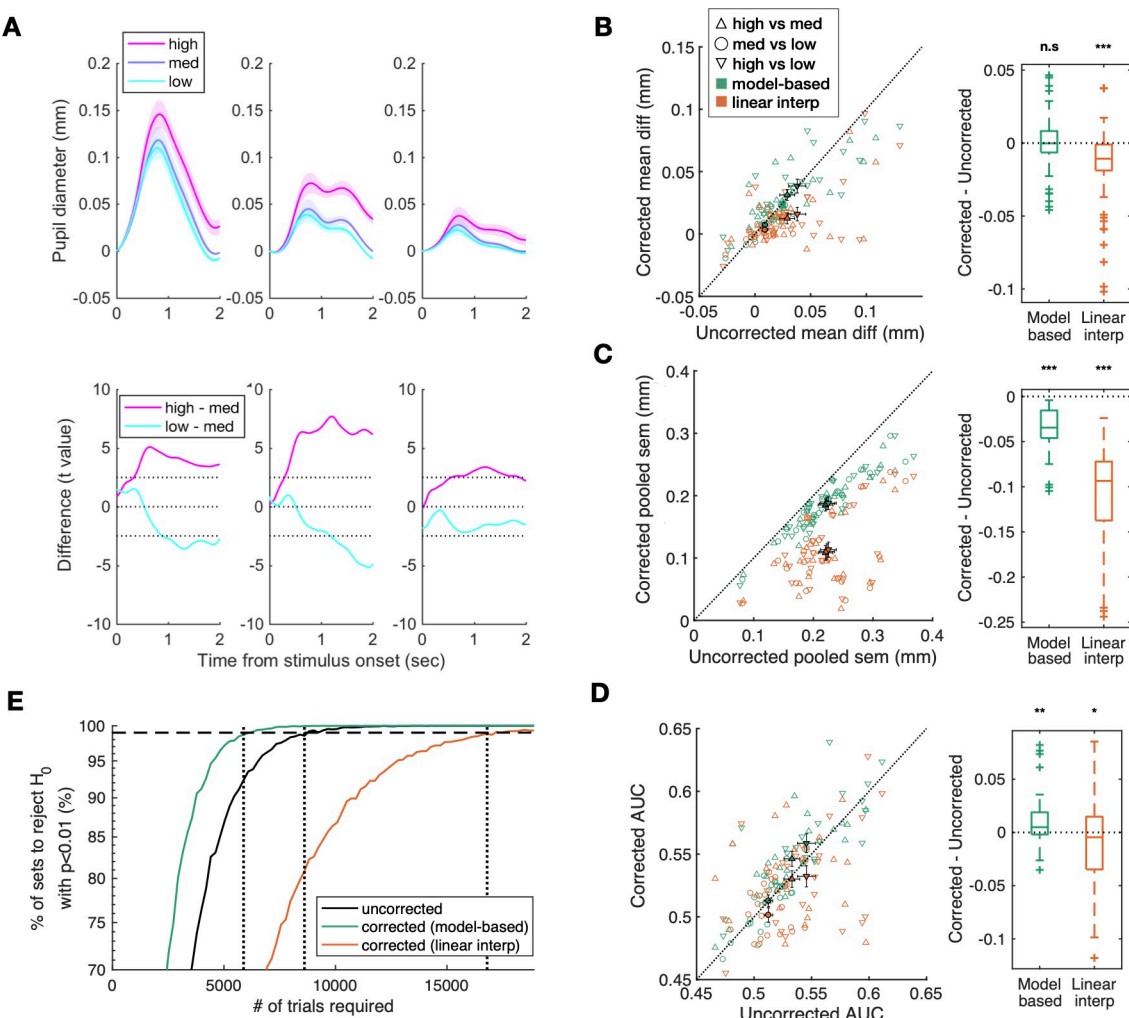

**Fig 7. Contribution of BPR correction to the statistical power of the auditory oddball task.** (**A**) (top panels) Time courses of pupil size for different levels of sound-frequency oddity. Lines and shades correspond to the means and SEMs across subjects. (bottom panels) Time courses of t statistics (paired t-tests, df = 23) for the deviations of the high oddity (magenta) and low oddity (cyan) conditions from the medium oddity (blue) conditions. The left panel shows the results before correction; the middle and right panels show the results after being corrected using the model-based method and the interpolation method, respectively. (**B-D**) (left panels) Diagonal plot. Each marker indicates each condition pair, a subject has three markers each (high vs medium, medium vs low, and high vs low). Filled marker and error bars indicate the mean and SEM across subjects. Empty markers correspond to individual subjects, with different colors indicating different correction methods (green for the model-based method; orange for the interpolation method) and different symbols indicating different oddity pairs (see inset for detailed labels). (right panels) Boxplot. Corrected–uncorrected, which are equivalents to deviance from the diagonal line in the corresponding left panels. (Wilcoxon signed-rank test, (*, $p < 0.05$; **, $p < 0.01$; ***, $p < 0.001$)). The color scheme matched that shown in the left panels. (**B**) Mean difference in pupil dilation, (**C**) pooled SEM, (**D**) AUC between the conditions. (**E**) Comparison of the data uncorrected (black line), the data corrected with the model-based method (green), and the data corrected with the linear interpolation method (orange) in statistical power. The fraction of significant tests was plotted against the number of trials (see Methods and Materials for the detailed procedure).

## Correcting delayed orientation estimation data for BPR

In line with previous work [19,29–31], the mean rate of eye blinking significantly fluctuated depending on the task epochs in the delayed orientation estimation task (black dotted line in Fig 8; see Materials and Methods for details about the task). The blink rate increased within each of the four task epochs, reaching the global maximum in the preparation epoch, a local

maximum right after the bar presentation, another local maximum right after masking stimulus presentation, and another at the start of the estimation epoch. The visual comparison of the average profile of the corrected pupillary responses (green line in Fig 8) with that of the pre-correction responses (black line in Fig 8) suggests that the correction algorithm selectively carved out BPR from the pre-correction pupillary responses in two aspects. First, the pupil size increased overall after correction, reflecting the baseline level of blink rate. Second, the first local dimple (bottom red arrow in Fig 8) in the preparation epoch (top red arrow in Fig 8), but not the second local dimple in the masking epoch, was removed after correction, which makes sense that the latter dimple is likely to reflect the pupil constriction due to the sudden increase of retinal illuminance at the beginning of the masking epoch.

Overall, the pupil began to dilate substantially right after stimulus offset and became maximal in amplitude during the estimation period (top blue arrow in Fig 8) regardless of whether the pupil data were corrected or not. This is anticipated from previous work on working memory, in which the pupil size increased as memoranda of stimuli need to be retrieved [5]. Additionally, the peak (as indicated by the vertical dashed lines in top panels of Fig 9A) increased as a function of the working-memory load (the number of bars to be remembered; Fig 9A, top). Notably, the blink rate also increased as the working-memory load increased, as shown before (the top panel of Fig 3D, which is redrawn in the top panels of Fig 9A). This implies that the

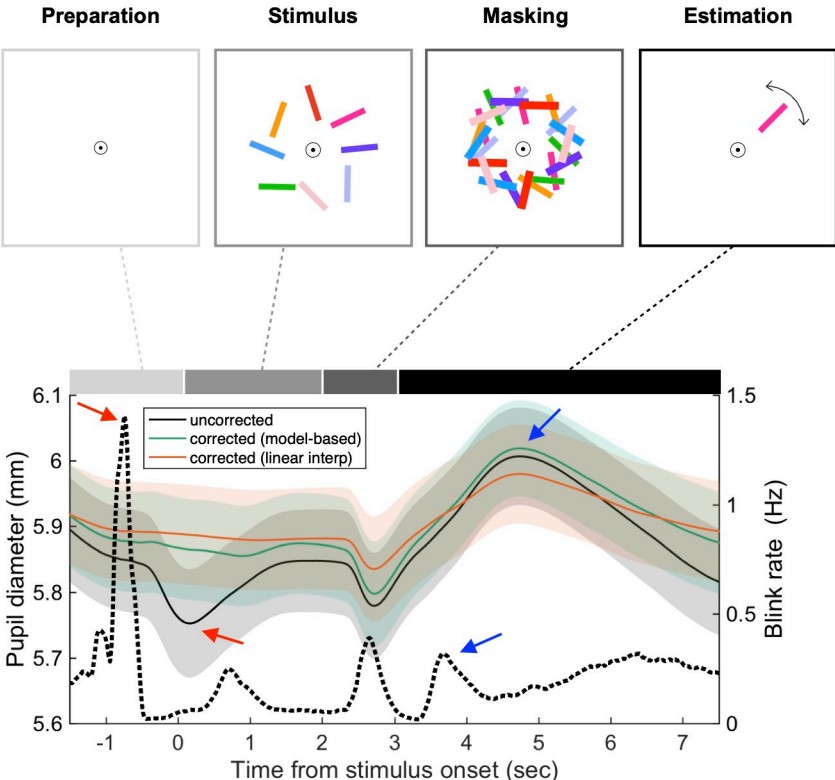

**Fig 8. Temporal bias in blink patterns induces spurious distortion of pupil time course in the delayed orientation estimation task.** Pupil (solid line) and blink rate (dotted line) time course during the delayed orientation estimation task. Since BPR induces pupillary constriction, the pupil-size time courses corrected by the model-based method (green) and the linear interpolation method (orange) were larger than pre-correction (black) overall, but note that the linear interpolation method decreased the time course when it was a hill (upper blue arrow). Note that subjects tend to blink more often at the preparation epoch, and it induces a spurious dip at stimulus onset (red arrows). It was flattened after applying both of the correction methods. Line and shade indicate the mean and SEM, respectively.

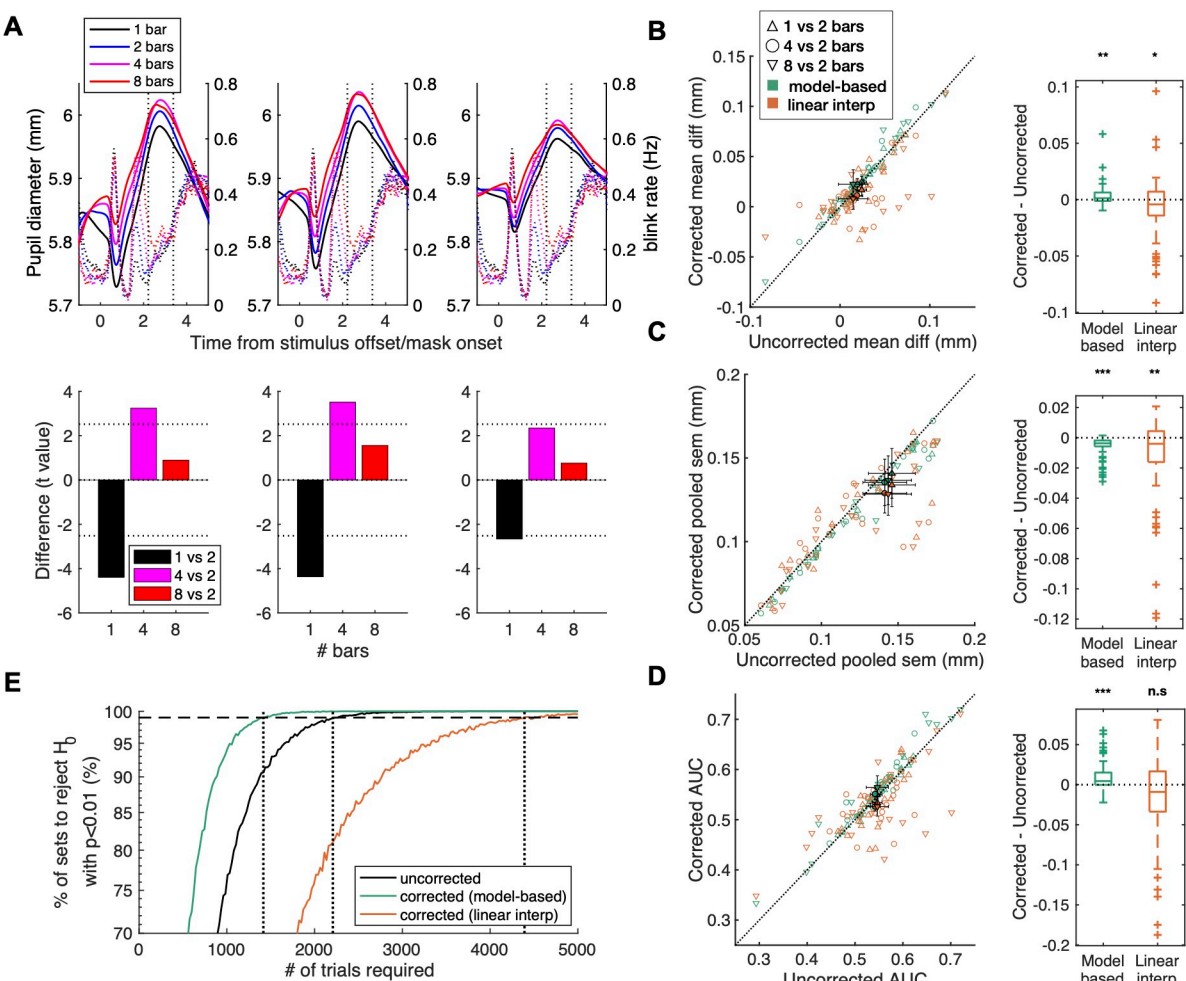

**Fig 9. Contribution of BPR correction to the statistical power of the delayed orientation estimation task.** (**A**) (top panels) Pupil (solid line) and blink rate (dotted line) time course across the working-memory load conditions. The time 0 is the onset of the mask epoch. The pupil size increased as a function of working-memory load. The color indicates the number of bars to be memorized (N = 1,2,4 and 8 are denoted as black, blue, magenta, and red lines, respectively). Solid lines indicate the means across subjects, and the vertical dotted lines indicate the time span where the mean pupil of each trial is computed. Note that the more memory load is given, the higher the blink rate was, and thereby BPR would be more biased to the higher memory load conditions. The model-based method captured this bias so that the pupil-size time course under high memory loads increased more than low memory loads after the correction. (bottom panels) T values of the paired t-tests (df = 20) for the difference of each bar condition from the 2-bar condition. The left panel shows the results before correction; the middle and right panels show the results after being corrected using the model-based method and the interpolation method, respectively. (**B-D**) (left panels) Diagonal plot. Each marker indicates each condition pair, a subject has three markers each (1 bar vs 2 bars, 4 bars vs 2 bars, and 8 bars vs 2 bars). Filled marker and error bars indicate the mean and SEM across subjects. Empty markers correspond to individual subjects, with different colors indicating different correction methods (green for the model-based method; orange for the interpolation method) and different symbols indicating different working-memory load (the number of bars) pairs (see inset for detailed labels). (right panels) Boxplot. Corrected–uncorrected, which are equivalents to deviance from the diagonal line in the corresponding left panels (Wilcoxon signed-rank test, (*, $p < 0.05$; **, $p < 0.01$; ***, $p < 0.001$)). The color scheme matched that shown in the left panels. (**B**) Mean difference in pupil dilation, (**C**) pooled SEM, (**D**) AUC between the conditions. Panel B and C were zoomed in for clear visualization. See S2 Fig for zoomed-out plots. (**E**) Effects of the correction methods on statistical power in the delayed orientation estimation task. The number of trials required to reject the null hypothesis decreased by about 40% by the model-based method, whereas it increased 100% by the linear interpolation method.

blink rate, via BPR, works as a serious confounder that counteracts the pupil-size changes associated with the cognitive state of interest, i.e., working-memory load. After being corrected for BPR with the model-based method, the differences between the working-memory load conditions increased (middle top panel of Fig 9A), resulting in increased statistical differences

between the conditions (middle bottom panel of Fig 9A). By contrast, the interpolation method rather substantially decreased the mean differences (right top panel of Fig 9A) and the statistical differences (right bottom panel of Fig 9A). These results showcase the effectiveness of our model-based method in selectively removing BPR while keeping the pupillary responses of interest intact.

Using the same procedure used for the auditory oddball task, we assessed the extent to which the correction algorithms contribute to the statistical power of revealing the differences in pupillary responses between the working-memory load conditions. We obtained the results that were similar to those of the auditory oddball task data: The discriminability between the working-memory load conditions across subjects increased after the BPR correction by the model-based method, whereas it is decreased after the correction by the interpolation method (Fig 9A bottom panels). Additionally, the model-based method enhanced the mean differences (green symbols in Fig 9B) and decreased the SEMs (green symbols in Fig 9C). As a result, the statistical power was significantly improved after correction (green symbols in Fig 9D; green line in Fig 9E). By contrast, the interpolation method decreased the statistical power (orange symbols in Fig 9D; orange line in Fig 9E) by substantially reducing the mean differences (orange symbols in Fig 9B).

## Discussion

### Summary of findings

By inspecting the pupillary response profiles that were associated with isolated single events of eye blinking, we learned that pupil size reacts to a single blink similar to how it responds to an abrupt change in retinal illuminance, which we dubbed BPR. We also learned that BPR varies substantially, mainly in amplitude and slightly in phase, across display luminance levels, subjects, and blinks. By inspecting how the blink rate changes as a function of task epoch or as a function of cognitive states, we also learned that BPR is not a mere nuisance but should be treated as a serious confounder that threatens the internal validity of experiments. Upon these empirical findings and understandings, we built a generative model in which a given pupillary measurement is the stochastic outcome of the linear summation of the cognitive-state, spontaneous-state, and blink variables, while the blink variable may confound pupillary measurements via the backdoor route involving BPR. With this generative model in hand, we argued that pupillary measurements should be corrected for BPR by counterfactually inferring BPR on a subject-to-subject and blink-to-blink basis, and developed an algorithm of doing such counterfactual inference. By analyzing the three data sets that were acquired using the fixation, auditory oddball, and delayed orientation estimation tasks, respectively, we showed that the generative model can explain away the seemingly peculiar event-locked profiles of the raw pupillary responses with the confounding acts of BPR. Lastly, we demonstrated that our newly proposed algorithm could effectively and selectively remove BPR and significantly increase the statistical power of revealing the pupillary-response differences between the cognitive states of interest. Our findings call for the attention of researchers who analyze pupillometry data to the presence of BPR, the serious nature of its confounding acts, and the proper way of addressing it.

### Potential causes of BPR

As mentioned briefly in INTRODUCTION, there are many reasons to believe that BPR is mainly caused by the transient changes in retinal input that occur as the eyelid briefly blocks the light pass through the pupil. First, previous work reported that "dark flashes"–turning off and on the light for a short period approximately matching the blink duration–under constant illumination make the pupil constrict and re-dilate with the dynamics close to BPR [15,41].

Specifically, successful description of the pupillary responses to dark flashes with a linear time-invariant system [41] suggests that the short-pipe shape of BPR dynamics can be accounted for by a linear mixture of slow and weak dynamics of dilation associated with the abrupt decrease in illuminance due to eye closing and a fast and strong dynamic of constriction associated with the abrupt increase in illuminance due to eye-opening. Second, saccadic eye movements also trigger the dynamics of pupil size that are similar to BPR [16,17,49,50]. Thus, pronounced pupil constriction followed by saccadic eye movements is similar to the pupillary responses to abrupt changes in visual input, such as a checkerboard undergoing polarity inversion. However, we also note that motor activity during blinks may also contribute to BPR, especially to the slow dilation component of BPR because a motor activity is known to promote pupil dilation [51].

## The shape and variability of BPR

As suggested by a few previous studies [15,41], the general profile of BPR identified in the current work resembled the known pupillary responses to a dark flash, which is probably caused by the abrupt change in retinal illuminance due to the eyelids' act of blocking the light entering the pupil. Furthermore, although a previous [17] study characterized the generic profile of BPR as a biphasic shape with a rebounding tail using the finite impulse response method, the rebound was largely absent or negligible in our data. Considering also that our correction algorithm, in which BPR was modeled as a single inverted gamma probability density function, did not exhibit any noticeable systematic bias, we think that a "short-pipe" shape can sufficiently describe BPR without a rebounding tail.

One novel contribution of the current work to understanding the properties of BPR is that the amplitude of BPR markedly varies across individual instances of eye blinking. For example, on average, the amplitude of BPR of the top 20 percentile was 7.85 times greater than that of the bottom 20 percentile. Although we have not specified the origin of this large variability of BPR amplitude, our studies strongly suggest that any correction methods based on the assumption of BPR with a constant amplitude are likely to leave substantial amounts of unwanted variability in the supposedly corrected pupillary measurements. This unaddressed variability may decrease the statistical power of the experiment and impose severe limits on any investigation of pupillary responses on a trial-to-trial basis, which provides crucial information about cognitive systems. In this regard, our correction algorithm can be considered one effective way of addressing the blink-by-blink variability of BPR amplitude.

## Confounding acts of BPR

One of the most important contributions of the current work to the field of pupillometry studies on cognitive processing is our demonstration that BPR might act as a serious confounder. We demonstrated two concrete examples of such confounding acts of BPR. When the trial length is shorter than the length of BPR, as in the auditory oddball task, the overarching influence of BPR, in conjunction with the practice of adapting the baseline to trial onset, can lead to spurious yet great amounts of distortion depending on when blinking events occur in relative to the current trial. We also demonstrated that when the cognitive state of interest is associated with the blink rate, as in the delayed orientation estimation task, the influence of the cognitive state on pupillary measurements is also distorted depending on the nature of such association. Considering that these two situations, i.e., short-event-related design and condition-dependent blink-rate modulation, are not rare cases in cognitive experiments [38,52], we argue that BPR should not be treated lightly as a mere nuisance but rather seriously as a confounder that would threaten the internal validity or statistical power of experiments.

Another important contribution of the current work is that we built a generative model of pupillary measurements. We stress that this generative model was built upon the knowledge, which was empirically earned from a set of controlled observations designed to characterize the key properties of BPR, such as its shape, variability, and confounding nature. By incorporating this empirical knowledge into the generative model in which the major variables shape a causal network of random variables, we conceptualized BPR as a backdoor route through which a confounding flow of information propagates [53]. One of many virtues of having a generative model is to allow for conducting ex-ante simulations, which allow for validating certain assumptions upon which the model is built. We exercised that virtue when validating the key assumption of the generative model regarding the aforementioned backdoor route (Fig 3) and when explaining away the seemingly peculiar profiles of pupillary measurements encountered in the observed data (Figs 5 and 8). To be sure, we do not claim that the generative model presented in the current work sufficiently includes the variables involved in pupillary measurements (e.g., a few obvious factors such as the light variable were omitted) nor that all assumptions have been empirically grounded (e.g., the linear summation of the outputs from the three variables or Independence of BPR from each blink). Our generative model should be considered a good approximation of how BPR relates to cognitive states, spontaneous state, and pupillary measurements. More future studies are required to validate further and extend this model.

## The proper way of correcting pupil-size measurements for BPR

We stress that our algorithm of correcting pupil-size measurements for BPR is a normative solution derived from the generative model. Given the causal structure of the variables in the generative model, the proper solution of correcting pupil-size measurements for BPR must be to selectively block the backdoor route from the cognitive state to the pupil size via the blink variable. Put in terms of the Bayesian network, the proposed solution can be understood as the strategy of "counterfactually" fixing the blink variable at the "off" state although the observed (actual) state of the blink variable is "on." Such fixing requires the correct specification of the causal function from the blink variable to pupil-size measurements, BPR. From the fixation task, we learned the three crucial features of this causal function: (i) in general, BPR has a 3-s-long temporal profile of fast constriction followed by slow re-dilation; (ii) individual subjects have their own unique, and fixed shape of BPR; (iii) for a given individual, the amplitude of BPR stochastically varies on a blink-to-blink basis. This means that the shape and amplitude of BPR must be "probabilistically inferred" from the observed pupil-size measurements and the prior knowledge about BPR. Consistent with this implication, we showed that the pupil-size measurements could not be corrected for BPR with conventional approaches such as averaging a substantially large number of trials or replacing the blink-affected epoch of measurements with an interpolated line. One may wonder whether this blink-to-blink probabilistic inference of the shape and amplitude of BPR can be approximated by applying interpolation-based methods, including sophisticated versions that perform correction in a blink-to-blink manner such as "cili" [54]. However, such interpolation-based methods are limited because the interpolation procedure not just removes BPR but also throws away the genuine pupil-size changes of interest, i.e., pupillary responses to certain cognitive factors. Because of this limitation, the interpolation-based correction of BPR might end up decreasing the statistical power of experiments, as we demonstrated in our results.

To our best knowledge, there has been only one previous work that attempts to correct pupil-size measurements for BPR by estimating the shape and amplitude of BPR [17]. However, this approach differs from ours in two crucial aspects. First, it is assumed that the shape

of BPR might be unique to a given individual, as we did, but the amplitude of BPR was fixed at a constant value. Secondly, it estimated the shape of BPR using the finite impulse response method, which requires the design matrix of events because the pupillary measurements were modeled as the sum of pupillary responses to events. As mentioned above, the assumption of the fixed amplitude of BPR will lead to substantial inflation of blink-by-blink variability, given the high degree of variability of BPR amplitude found in the current work. In addition, unlike this approach, our correction algorithm does not require any design matrix of events and thus can be applied to a wide range of experimental designs.

Although our correction algorithm is grounded in the empirically constrained generative model, effective in selectively filtering out BPR, and contributes to the statistical power, a few of its limitations should be mentioned. First, our algorithm may require a large volume of data that is sufficient to build a reliable probability distribution because it estimates the amplitude of BPR on a blink-to-blink basis based on the probability distribution. We think that the volume of data can be compromised by assuming the constant-amplitude BPR in specific experimental designs, whereby the trial-to-trial variability is not important, or by using the canonical shape of BPR when individual differences do not matter much. Second, in principle, it would be difficult to distinguish between BPR and the pupillary responses associated with the cognitive factor of interest when the latter's shape and timing are close to those of BPR. However, we stress that it is very rare for the pupil time course of a certain cognitive event to match BPR exactly in timing and shape. Lastly, our algorithm relies on the fitting procedure that is rather slow, computationally costly, and still imprecise, especially when blinks occur too frequently or in a burst-like manner. However, these problems are not specific to our algorithm but rather general to all model-based correction methods.

To help readers readily apply the BRP-correction algorithm developed in the current work to their own data, we provided it as an out-of-the-box solution. This toolbox is coded in MATLAB and requires little manual editing to work: all users have to do is feed in time series of pupil-size measurements and blink data as inputs to the toolbox. The MATLAB codes, prerequisite data-preprocessing steps, and a brief description of the correction procedure with an example data set can be downloaded at the following link: https://github.com/yookyung1310/BPR_toolbox.

## Practical considerations for avoiding BPR-related problems

Given the prevalence of spontaneous blinks and the confounding nature of BPR, we considered a few points that pupillometry experimenters should keep in mind at the various stages of conducting experiments.

At the experimental design stage, consider making the inter-trial interval sufficiently long, at least more than 3 s. This recommendation is worth being considered even when blinks do not occur because the pupil size needs some time to return to its baseline size. However, the presence of BPR even further recommends researchers to make the inter-trial interval long, especially given that subjects tend to blink a lot at implicit breakpoints such as the moment right after making responses. Thus, the separation of consecutive trials by more than 3 s will tend to minimize–although it will not prevent completely–the influences of BPR on raw pupil-size measurements.

At the data acquisition stage, it is recommended to record a whole time series of eye-tracking data over an entire run. Suppose data are acquired separately for individual trials and baselined to the pupil-size measurement at trial onset. In that case, there is no way to correct these baselined time courses of measurements for BPR. Alternatively, even when data are collected in a trial-by-trial manner, we recommend experimenters to keep the time series of at least 3 s

before the trial onset so that BPR can be delineated before adapting the data to a baseline level. Note that this practice is worth being considered also because it allows for measuring the pupil size before any experimental manipulation of interest.

At the stage of giving subjects instructions regarding task performance, we recommend that experimenters consider not asking subjects to suppress spontaneous blinks for several reasons, despite the existence of BPR. First, the blink rate is highly informative, known as a correlate of many important neuro-cognitive states, including reward, attention, and dopamine activity [32–38]. Intentional suppression of blinks would distort such valuable information. Second, the instruction of blink suppression will aggravate eye fatigue from eye-tracking experiments. Third, in a typical video-based pupillometry setup, being a subject is already quite demanding: the eyes being exposed to the infrared light; the gaze direction is fixed at the fixation target; the body posture is maintained at a specific position which is best for data collection for experimenters but might be uncomfortable for subjects. Thus, adding another request of suppressing blinks on top of an already straining set of requests will likely make subjects weary and lose focus on task performance. Lastly, an attempt to suppress blinks for a prolonged period may induce unwanted cognitive and neural responses such as the accumulation of a natural bodily urge [39].

At the stage of data preprocessing, we do not recommend experimenters discard blink-affected trials. The selective removal of blink-affected trials reduces the total number of trials for data analysis and is likely to invite another confounding factor, especially when the blink rate is tightly linked with a cognitive state of interest. For example, suppose the true time course of pupil size differs between blink-affected and blink-free trials, possibly due to the difference in dopamine activity or working-memory load, for instance. In such cases, deleting only the blink-affected trials would bias results such that the underlying difference between blink-affected and blink-free trials is not taken into account in the observed differences between experimental conditions.

## Materials and methods

### Subjects

Paid 28 (21 females, aged 23.7 ± 3.3), 27 (19 females, 25.6±3.6), and 22 (12 females, aged 26.4 ±3.3) human subjects participated in the fixation, auditory oddball, and delayed orientation estimation tasks, respectively. Thirteen subjects participated in more than one experiment: 11 in all the three experiments; 2 in the fixation and auditory oddball experiments. All had normal or corrected-to-normal vision with contact lenses or glasses. Subjects were recommended not to wear glasses because the light reflected from glasses often occludes the pupil. The Research Ethics Committee of Seoul National University approved the experimental procedures (IRB No. 2007/003-029). All participants gave informed consent before the experiments and were naïve to the purpose of the experiments. The data quality was not good when the eyelids occluded the pupil substantively, the blink rate was too high (> 40/mins), or subjects dozed off during data collection. As result, one, three, and one subjects were excluded from data analysis for the fixation, auditory oddball, and delayed orientation estimation tasks, respectively. However, we did not exclude trials or subjects based on task performance because task performance was generally good for all the participants. In both of the auditory oddball and delayed estimation tasks, subjects in our study showed a reasonable range of performance in accuracy (mean ±SD = 97.74±2.59%, range = 90.4%-99.9% for the auditory oddball experiment; absolute error mean±SD = 6.19±1.74, 8.93±3.06, 12.94±4.43, 21.89±5.88 deg, range = 4.35–10.90, 4.86–18.22, 8.01–23.03, 15.89–35.11 deg for each memory load of the delayed estimation experiment) and RT (mean±SD = 360±63 msec, range = 201–454 msec for the auditory oddball experiment;

mean±SD = 1.95±0.42, 2.02±0.49, 2.17±0.55, 2.31±0.62, range = 1.45–3.24, 1.46–3.70, 1.58–4.09, 1.67–4.35 sec for each memory load of the delayed estimation experiment), which were quite comparable to those reported in the previous work using the same tasks [44,47,55].

## Apparatus and eye-tracking setup

All experiments were conducted in a scotopic room. Subjects' heads were positioned on a chin rest in front of a monitor (LG FLATRON 19-inch monitor for the fixation task; Dell 4k monitor for the remaining two tasks). The distance between the chin rest and the monitor was 60 cm (in the fixation task) or 90cm (in the remaining two tasks). Pupil diameters and gaze positions were binocularly recorded with the Eyelink 1000 plus in the fixation task or the Eyelink 1000 system (SR Research) in the remaining two tasks at a sampling rate of 500Hz. The eye tracker was calibrated using the built-in five-point calibration routine (HV5) at the beginning of each experimental session, after each break, whenever the background luminance changed, or occasionally when the subjects' head positions were found to move from the original position during runs. Eye-tracking signals were acquired in a 'pupil–corneal reflection (P-CR)' mode. The pupil size was estimated using the built-in ellipsoid fitting method, which is known to be robust for pupil occlusion by the eyelids.

## Stimuli and procedure

The stimuli were generated using MATLAB (MathWorks) in conjunction with Psychtoolbox-3 [56–58]. To prevent any unwanted processes associated with voluntary blink suppression, such as a natural body urge [39], we intentionally did not give the subject any instructions about eye blinking at all throughout the entire experimental sessions. Subjects were allowed to take as many breaks as desired at the end of each run, disengaging from the eye-tracking setup and moisturizing the eyes using disposable artificial tears as needed. Experimenters monitored the state of subjects to check whether they fall asleep or have dry eyes. Subjects were recommended to take a break or nap for a few minutes if they felt sleepy.

**Fixation task.**   Each subject participated in three sessions, each of which consisted of twelve 140-s-long eye-tracking runs. To check whether and how the shape and amplitude of BPR change as a function of baseline pupil size, we varied the background luminance of the display with four levels. To do so, the lowest and highest possible levels of luminance were included, and a black strawboard was attached to the monitor to block the light thoroughly for the darkest background-luminance condition. The two intermediate levels were chosen such that the corneal flux density (retinal illuminance), a product of visual area and luminance from the area, which determines baseline pupil size [42], fell in a dynamic range. As a result, the background-luminance levels were 0.00, 0.03, 5.35, and 97.60 cd/m$^2$, which corresponded to 7.8, 62.3, 5256.8, and 95790.0 cd/m$^2$/deg$^2$ of corneal flux density in the current experimental setup with fixation stimuli and the infrared light from the eye tracker. Each background-luminance run was repeated in a row over three runs. The order of the highest three background-luminance conditions was determined using the Latin square method, while the lowest background-luminance condition was always assigned to the last three runs because the strawboard had to be attached unlike the other conditions. Subjects were instructed to fixate on a fixation cross (height/width: 1.47 visual angular degrees (v.a.d.)). The luminance of the fixation cross was 97.6 cd/m$^2$ for the lowest two background-luminance conditions and 0.03 cd/m$^2$ for the two highest conditions for visibility. However, note that both the luminance of the fixation cross and the background always remain unchanged within each eye-tracking run. Gaze-related trial exclusion was not applied.

**Auditory oddball task.** Two brief sine-tone auditory stimuli with different frequencies (200 ms in length, 1.5 kHz and 1 kHz) were heard via external speakers. To help subjects remain attentive and prevent unwanted pupil-size measurement errors due to visual gaze, we instructed subjects to fixate at a bull's eye stimulus (outer radius: 0.15 v.a.d., inner radius: 0.03 v.a.d., 0.06 cd/m$^2$) on the darkest background-luminance screen with a large circle (18 v.a.d., 31.10 cd/m$^2$). To measure the changes in pupil size associated with stimulus oddity while preventing those associated with motor responses or mental counting, we asked subjects to perform a two-alternative forced-choice (2AFC) task so that a button press was made every trial. Subjects were instructed to respond as fast and accurately as possible by pressing the"1" or"2" key of a number pad to low-tone or high-tone stimuli while fixating to the fixation target throughout entire runs. The tone-key assignment was counterbalanced across subjects. There was no feedback for performance. There were three types of runs: (i) in the 'high-tone-odd' run, the high-tone and low-tone stimuli were presented for 20% and 80%, respectively, of trials; (ii) in the 'low-tone-odd' run, the high-tone and low-tone stimuli were presented for 80% and 20%, respectively, of trials; (iii) in the 'no-odd' run, the high-tone and low-tone stimuli were presented for 50% and 50%, respectively, of trials. A single run consisted of 70 trials, and subjects participated in 14 to 18 runs (14 runs for one subject; 15 runs for 19 subjects; 18 runs for 4 subjects). The sequence of stimuli was pseudo-randomized. The inter-stimulus interval was fixed at 2 s. Gaze-related trial exclusion was not applied. Deviance from bull's eye was 1.79 ± 0.94 deg (mean ± SD across subjects)

**Delayed orientation estimation task.** Subjects were instructed to fixate their gaze at a bull's-eye fixation stimulus (outer radius: 0.15 v.a.d., inner radius: 0.03 v.a.d., 57.10 cd/m$^2$) on the center of the gray (27.72 cd/m$^2$) background. Each trial consisted of six epochs, preparation (0.5–1 s), stimulus presentation (2 s), masking (1 s), orientation estimation (up to 6 sec), confidence reporting (no time limit), and waiting (1 s) epochs. Note that the duration of trial varied because the length of the orientation estimation (2.11 ± 0.51 sec (mean ± SD across subjects, min = 1.55, max = 3.85 sec)) and confidence reporting epochs (0.77 ± 0.29 sec (mean ± SD across subjects, min = 0.41, max = 1.54 sec)) depend on how quickly subjects respond. In the preparation epoch, by acquiring subjects' gaze data online, we aborted trials in which the gaze deviated more than 2 v.a.d. from the fixation center and re-administered those trials later. This abortion procedure was omitted in some runs, where online gaze data were unstable. In the stimulus presentation and masking epochs, oriented bar stimuli with different colors, the number of which varied from 1 to 2, 4, and 8 bars, were presented and followed by masking stimuli, which consisted of 90 bars that changed their color and orientation every video frame (60 Hz). In the orientation estimation epoch, subjects had to estimate the post-cued (by color) orientation of the presented bars by rotating a probe bar stimulus as close to its original orientation as possible. In the confidence reporting epoch, subjects reported their subjective confidence about orientation estimation with a slider. The sequence of bar number, orientation and location was pseudo-randomized over trials to minimize any order-related confounding effects. Each run consisted of 48 trials. Mean estimation errors (in angular degrees) and mean confidence was shown as feedback for overall performance at the end of each run. Subjects conducted 8 runs in a single daily session (for 11 subjects) or 20 runs in 2 or 5 daily sessions (for 10 subjects).

## Data analysis

**Preprocessing of eye-tracking data.** The data from the eye tracker, which was provided in a digitized format called EDF, were imported to MATLAB using an open-source script (https://github.com/iandol/opticka/blob/master/communication/edfmex.m) and analyzed using the

custom MATLAB scripts. In all experiments, pupil size and gaze positions were measured binocularly. However, we opted to use the data from one of the two eyes that provided more reliable data for the following reasons. First, there were subtle but noticeable differences in blink offset timing between the two eyes. Because we wanted to define the time course of BPR precisely, we were concerned that averaging two signals that differ in timing might introduce unwanted blurs. Second, occasionally in some individuals, only one eye's data was not detected or became unstable probably due to eyelid occlusion, which is known to occur more frequently in Asian people. If we average the signals from the two eyes, the time courses are expected to show weak but spurious fluctuations due to the missed signals from the unreliable eye.

**The definition of single blink events.** Blink events were defined in the following procedure. Initially, we identified the time points where pupil data were missing, extremely small (<1 mm), or underwent abrupt changes. Here, to identify such "abrupt changes," we high-passed the data using the 3rd order Butterworth filter with 10 Hz cutoff frequency and detected the time points where the absolute values of the high-passed data exceeded 0.25 mm. Next, by assuming that the consecutive eye blinks are unlikely to occur in a row within 200 ms, we treated the initial time points that were apart less than 200 ms as belonging to a single blink event. The start and end time points of this "single blink event" were defined as the onset and offset of an individual eye blink. The mean blink duration defined in the above procedure was 288 ± 108 ms (mean ± SD across subjects) for the fixation task, 311 ± 82 ms for the auditory oddball task, and 349 ± 107 ms for the delayed orientation estimation task. The mean blink rate defined in the above procedure was 0.28 ± 0.12 Hz (mean ± SD across subjects) for the fixation task, 0.39 ± 0.18 Hz for the auditory oddball task, 0.28 ± 0.11 Hz for the delayed orientation estimation task.

**The removal of artifacts around single blink events.** It is known that the video-based eye-tracking data are typically contaminated with artifacts immediately before and after each blink event. For this reason, previous studies adopted a custom of getting rid of 150 ms around single blink events, i.e., replacing the data of those time points with not-a-number (NaN) values [59–61]. We also applied the same procedure for the time points before each blink but slightly modified the procedure for those after each blink, as follows, because the post-blink artifacts appeared more prolonged compared to the pre-blink artifacts. First, we computed the derivatives of the raw data and smoothed them by taking their moving averages with a boxcar time window (0.25 s in size). The smoothing procedure was applied not to be oversensitive for detection in the following step. Next, we determined the endpoint of the post-blink artifact by identifying the time point where the smoothed derivative starts to be smaller than 0.1 mm/s. We also constrained that the post-blink artifact period is longer than 200 ms and smaller than 500 ms. As the final step, the pre-blink and post-blink artifact time points were replaced with NaN values and linearly interpolated. Note that the smoothing with a boxcar window was applied only when defining the endpoint of the post-blink artifact, but not when preprocessing the pupil time courses that were used for analysis. Also, we note that a recent work [62] provided a more advanced algorithm to define blink events and to avoid pre-blink and post-blink artifacts. We also stress that this procedure only removes the artifacts that occur within a short period around blink events and should be distinguished from the procedure of correcting the pupil data for BPR, which is provided in the current work.

After interpolating the blink events and associated artifact time points, the pupil time courses were band-pass filtered (0.02 and 4Hz) with the 3rd order Butterworth filter to remove electric noise and slow drift components. Then the data, which was originally provided in an arbitrary unit, was converted into square millimeters by measuring the relationship between the original arbitrary unit and physical square millimeters. In doing so, we printed black dots, and then measured the size of the dots with a pupil-only mode (Eyelink 1000), or with

pinching holes into the dots and attaching a silver foil underneath the holes (Eyelink 1000 plus). The converted square-millimeter unit was re-converted into millimeters for data analysis as follows: $diamteter = \sqrt{Area/\pi}$, assuming that the shape of the pupil is a perfect circle.

**Analyses on gaze deviance and potential dependency between blink and gaze.** Since it is known that large changes in gaze position can distort pupil-size measurements by tilting eye images [63], we checked the extent to which the actual gaze position deviated from the fixation target. The amount of gaze deviation was 1.45 ± 0.09, 1.79 ± 0.94, and 1.84 ± 0.64 v.a.d. (mean ± SD across subjects), and 0.52 ± 0.23, 0.75 ± 0.40, 0.94 ± 0.38 v.a.d. (mean ± SD of the within-subject SDs) for the fixation, the auditory oddball, the delayed orientation estimation tasks, respectively. Although these amounts of deviances can be considered small, there might be some noise from gaze deviance in our pupil data. Thus, we checked the possibility that the potential pupil change confounded by gaze deviance could have affected our claims about the effect of blinks on pupil-size measurements. We reasoned that our claim is threatened only when blinks are correlated, either positively or negatively, with the amount of gaze deviances. However, we found that the cross-correlation between blink events and gaze deviances was close to zero or almost negligible (maximum cross correlation < 0.025 in any runs, in both horizontal and vertical gaze deviances). Based on these results, we concluded that the pupil changes due to gaze deviances might have added some noise to our data but are unlikely to affect our findings regarding the relationship between blinks and pupillary responses because blinks were not correlated with pupil-size measurements.

**Definition of isolated blink-affected and blink-free time courses.** To characterize the shape of BPR that is associated with a single, isolated blink in the fixation task, we identified only the blinks that are apart from neighboring blinks by more than 3 s. To ensure the reliability of data, we included only the data from the subjects for whom more than 25 isolated blinks were identified for each background-luminance condition. As a result, the data from 21 subjects contributed to the definition of isolated blink-affected time courses (Fig 2C–2F). To define the blink-free time courses, we took the following steps. First, we identified only the inter-blink intervals with more than 6 s. Next, we removed the initial 3-s portion, which is not free from a blink. Then, we divided the remaining portion into as many 3-s portions as possible (e.g., if the remaining portion is 7-s long, then we obtained two segments of blink-free time courses and discarded the 1-s portion at the end). The grand average of these blink-free time courses is shown as the blue curve in Fig 4C.

**Analysis of the blink-by-blink variability of BPR in peak amplitude and time-to-the-peak.** By inspecting the grand average of blink-affected time courses across subjects and background-luminance conditions, we found that it reached the negative peak at around 0.5–1.2 s after blink offset (Fig 2C). Based on this finding, we quantified the amplitude of individual BPRs by taking the lowest pupil size within the period of 0.5–1.2 s after blink offset. Next, for each subject, these individual BPRs were separately grouped according to the four background-luminance levels and then sorted by their amplitudes into five quantile bins within each background-luminance group. Then, for each quantile, BPR time courses were averaged across the background-luminance groups such that the BPRs from the 4 luminance levels equally contributed to the averaged time course. For reliable estimation of the peak amplitude and time-to-the-peak, these averaged time courses were further corrected for noise with a cubic splicing method. Lastly, for each quantile, we defined the peak amplitude and the time-to-the-peak by finding the negative maximum pupil size and the time point when that maximum pupil size was found, respectively (Fig 2F).

**Prediction of BPR confound via simulation.** To illustratively demonstrate how BPR can affect the pupil time courses and lead to incorrect interpretation of the pupil data acquired in

cognitive experiments, we conducted *ex-ante* simulations using the BPR learned from the fixation task on both the auditory oddball and the delayed orientation estimation tasks. The detailed procedure was as follows. First, the BPR that was defined at the second-highest background-luminance condition in the fixation task (the second brightest curve in Fig 2D) was used as the model BPR because the retinal illuminance under that condition was closest to the background-luminance level used in the auditory oddball and the delayed orientation estimation tasks. Second, to demonstrate the pure impact of BPR, we convoluted with this model BPR the time courses of blink rates that were empirically acquired in the auditory oddball and the delayed orientation estimation tasks (Fig 3B and 3D). Next, to demonstrate the impact of BPR in the presence of a certain pupillary response that is associated with a cognitive process of interest (i.e., responses to the sound and the manual response to it), we initially convoluted the blink rate from the auditory oddball task with the model BPR (as was done in the previous step) and add this BPR-convoluted time course (the red curve in Fig 5B) to the presumed pupillary response to a cognitive event (the blue curve in Fig 5B). In doing so, to specifically demonstrate how the BPR spuriously distorts the pupil responses to a cognitive event of interest, we contrasted the simulated responses associated with two specific sequences of trials, namely the "blink-in-current-trial" sequence and the "blink-in-previous-trial" sequence. These two sequences were both defined by the series of blink events in current and 1, 2, and 3-back trials ([t-3,t-2,t-1,t], where t stands for a current trial). To isolate the impact of BPR that occurred in previous (t-1) and current (t), both the two sequences were constrained for cases where blink events did not occur on t-2 and t-3 trials. Thus, the trial series of blink events were [0 0 0 1] for the "blink-in-current-trial" sequence and [0 0 1 0] for the "blink-in-previous-trial" sequence, where 0 and 1 stands for the absence and presence of blink events, respectively.

**Analysis of the blink rate and pupil data in the auditory oddball and delayed orientation estimation tasks.** To quantify how the frequency of blinks fluctuates over time within a trial, we computed the blink rate by taking the moving (boxcar, size = 0.1 sec) averages of blink offset pulses across trials (as shown in the top panels of Fig 3B and 3D and dotted line in the bottom panel of Figs 8 and 9A). To control for the differences in overall pupil size across runs, the pupil time courses were demeaned run-by-run for the mean difference and pooled SEM analyses, or z-scored run by run for the AUC and bootstrapping analysis. We demeaned the mean difference and pooled SEM analyses because z-scoring applies different denominators across runs, so cannot compare the change in mean or SEM directly. Next, trial-locked pupil time courses were baselined at the start of each trial. To quantify the magnitude of those trial-locked pupil responses that were associated with cognitive events of interest (i.e., "hearing expected or unexpected sounds" and "retrieval of a bar orientation with different working-memory loads"), we averaged the data within certain time windows. In the auditory oddball task, the start point of the time window was set to 0.5 after event onset because it appears to take at least 0.5 s for the pupil size to respond to events. The endpoint of the time window was set to the endpoint of the trial (2 s). In the delayed orientation estimation task, we used the time points where the grand average of pupil-size measurements across memory load conditions were above 75% of its maximum as the start and endpoint of the integration window (1.22–2.38 s). Having quantified these response magnitudes for each trial, we sorted them into the experimental conditions and computed their mean and its standard error for each condition and each subject, which was used to plot the data in Figs 7B, 7C, 9B and 9C. To evaluate whether and how much the BPR correction methods improve the discriminability in pupil size between conditions, we derived the receiver operation characteristic (ROC) curve from a given pair of distributions of trial-to-trial response magnitudes and computed the area-under-the-ROC curve for each pair of conditions, as shown in Figs 7D and 9D.

**Statistical power analysis.** To evaluate how much the BPR-correction algorithms contribute to the statistical power of experiments, we carried out the regression analysis on the bootstrap samples drawn from the original data of the auditory oddball and the delayed orientation estimation tasks while varying the bootstrap sample size. First, for a given sample size, bootstrap samples were repeatedly (10,000 times) drawn with replacement from the original pool of trial-to-trial response magnitudes, which were merged across subjects. Then, we linearly regressed the data onto the oddity levels (for the auditory oddball task) or the working-memory loads (for the delayed orientation estimation task). Specifically, the tones with 80%, 50%, and 20% probability were coded as -0.5, 0, and 0.5, respectively, as the regressor for the auditory oddball task; the 1, 2, 4, 8 bar conditions were coded as 1, 2, 3, and 4, respectively, as the regressor for the delayed orientation estimation task. The linear regression was implemented with the "fitlm.m" function in the Statistical and Machine Learning Toolbox of MATLAB. Lastly, we plotted the percentage of the bootstrap samples in which the regression was statistically significant ($p < 0.01$) as a function of sample size and computed how many trials are needed to reach the null hypothesis rejection fraction of 95% for the BPR-uncorrected data and the BPR-corrected data (Figs 7E and 9E).

## The algorithm of correcting pupillary responses for BPR

**The generative model.** At the core of our BPR correction algorithm lies the probabilistic inference about the generic shape of BPR, $\boldsymbol{h}$, and its blink-to-blink amplitude, $\boldsymbol{\theta}_j$ (where $j$ stands for $j$th blink). Based on what we learned from the fixation task, we built the generative model for this inference, which consisted of two parts, one for the "blink-free" time courses and the other for the "blink-affected" time courses (see the subsection titled "**Definition of blink-affected and blink-free time courses**" in the above for the detailed definition of these two types of times courses). The original, 3-s-long, time courses were down-sampled from 500 Hz to 5 Hz to minimize computing load, which resulted in vectors of 16 ({0 s, 0.2 s, 0.4 s, . . ., 3 s}) × 1 dimensions.

We modeled the blink-free time courses collected at the $i$th sample under the lighting condition $c$ from the subject $s$, $\mathbf{Y}_{isc}^{free}$, as the linear sum of the time course of ongoing spontaneous fluctuation, $\mathbf{SF}_{isc}$, and that of responses associated with a cognitive state of interest, $\mathbf{PRC}_{isc}^{free}$:

$$\mathbf{Y}_{isc}^{free} = \mathbf{SF}_{isc} + \mathbf{PRC}_{isc}^{free} \tag{Eq 1a}$$

Note that $s$ and $c$ are used simply to indicate that the algorithm should be applied separately to the data collected from a particular individual under a particular lighting condition.

Next, we assumed that the power transform of $\mathbf{Y}_{isc}^{free}$ is a stochastic sample drawn from a multivariate normal distribution with the mean, $\boldsymbol{\mu}_{sc}^{free}$, and the covariance, $\boldsymbol{\Sigma}_{sc}$:

$$g(\mathbf{Y}_{isc}^{free}; \lambda_{sc}) \sim MVN(\boldsymbol{\mu}_{sc}^{free}, \boldsymbol{\Sigma}_{sc}) \tag{Eq 1b}$$

where $g$ is the box-cox power transformation with a free power parameter, $\lambda$. Here, $\boldsymbol{\mu}_{sc}^{free}$ can be approximated with the mean time course of the blink-free time courses for the $c$ lighting condition from the subject $s$, $\hat{\boldsymbol{\mu}}_{sc}^{free}$. As current pupil sizes are strongly correlated with neighboring values, $\boldsymbol{\Sigma}_{sc}$ was approximated as the first-order autoregressive function with two parameters, $\sigma_{sc}$ and $\rho_{sc}$:

$$(\hat{\boldsymbol{\Sigma}}_{sc})_{k,l} = \sigma_{sc}^2 \rho_{sc}^{|k-l|} (k, l = 1, 2, 3, \ldots, 16) \tag{Eq 1c}$$

In sum, Eq 1a–1c describes the generative process for the blink-free samples with three parameters, $\{\lambda_{ssc}, \sigma_{sc}, \rho_{sc}\}$.

Next, the 3-s-long, individual time courses that were affected by $j$th blink, $\mathbf{Y}_{jsc}^{affected}$, were modeled as the linear sum of the time course of ongoing spontaneous fluctuation, $\mathbf{SF}_{jsc}$, that of

responses associated with a cognitive state of interest, $\mathbf{PRC}_{jsc}^{affected}$, and that of BPR responses, $\mathbf{BPR}_{jsc}$:

$$\mathbf{Y}_{jsc}^{affected} = \mathbf{BPR}_{jsc} + \mathbf{SF}_{jsc} + \mathbf{PRC}_{jsc}^{affected} \tag{Eq 2a}$$

where, based on what was learned in the fixation task, $\mathbf{BPR}_{jsc}$ was assumed to be constant in shape under a given lighting condition for a given subject, $\mathbf{h}_{sc}$, but vary in amplitude from blink to blink, $\boldsymbol{\theta}_{jsc}$:

$$\mathbf{BPR}_{jsc} = \theta_{jsc} * \mathbf{h}_{sc} \tag{Eq 2b}$$

where $\mathbf{h}_{sc}$ was modeled as a gamma probability density function, $f$, with two shape parameters, $\alpha$ and $\beta$, one amplitude parameter, $\gamma$, and one temporal shift parameter, t, as follows:

$$\mathbf{h}_{sc}(\mathbf{x}; \alpha_{sc}, \beta_{sc}, \gamma_{sc}, t_{sc}) = \gamma_{sc} * f(\mathbf{x} - t_{sc}; \alpha_{sc}, \beta_{sc}) \tag{Eq 2c}$$

where $\mathbf{x}$ is a time vector comprising the blink-affected time course. As was done for the blink-free time courses, the power transform of $\mathbf{Y}_{jsc}^{affected} - \mathbf{BPR}_{jsc}$ was assumed to be a stochastic sample drawn from a multivariate normal distribution with the mean, $\boldsymbol{\mu}_{sc}^{affected}$, and the covariance, $\boldsymbol{\Sigma}_{sc}$:

$$g(\mathbf{Y}_{jsc}^{affected} - \mathbf{BPR}_{jsc}; \lambda_{sc}) = g(\mathbf{SF}_{jsc} + \mathbf{PRC}_{jsc}^{affected}; \lambda_{sc}) \sim MVN(\boldsymbol{\mu}_{sc}^{affected}, \boldsymbol{\Sigma}_{sc}) \tag{Eq 2d}$$

where $\boldsymbol{\mu}_{sc}^{affected}$ can be approximated with the cubic spline of both ends of the mean of the blink-affected time courses, $\hat{\boldsymbol{\mu}}_{sc}^{affected}$. In sum, Eq 2a–2d describes the generative process for the blink-affected samples with additional five parameters, $\{\theta_{jsc}, \alpha_{sc}, \beta_{sc}, \gamma_{sc}, t_{sc}\}$.

**Inference of generative model parameters.** Having defined the generative model as above, the goal of the BPR-correction algorithm becomes to infer the parameters of the generative models, a total of seven parameters for a given lighting condition for a given subject, $\{\lambda_{sc}, \sigma_{sc}, \rho_{sc}, \alpha_{sc}, \beta_{sc}, \gamma_{sc}, t_{sc}\}$ and 1 parameter for each blink $\theta_{jsc}$. These parameters can be inferred as follows. First, $\lambda_{sc}$ for the power transform can be estimated by the box-cox method. Second, $\sigma_{sc}$ and $\rho_{sc}$ for $\boldsymbol{\Sigma}_{sc}$ can be inferred by finding their values that maximize the log-likelihood of the observed $\mathbf{Y}_{isc}^{free}$:

$$\{\hat{\sigma}_{sc}, \hat{\rho}_{sc}\} = \underset{\sigma, \rho}{\operatorname{argmax}} \left[ \sum_{i=1}^{n_{free}} \ln(MVN(g(\mathbf{Y}_{isc}^{free}; \lambda_{sc}); \hat{\boldsymbol{\mu}}_{sc}^{free}, \hat{\boldsymbol{\Sigma}}_{sc})) \right] \tag{Eq 3a}$$

where $n_{free}$ is the number of blink-free samples. Third, $\{\alpha_{sc}, \beta_{sc}, \gamma_{sc}, t_{sc}\}$ for $\mathbf{h}_{sc}$ can be inferred by finding the values that maximize the likelihood of the power transform of the subtraction of BPR from the mean of all the blink-free samples, $\mathbf{Y}_{jsc}^{affected}$:

$$\{\hat{\alpha}_{sc}, \hat{\beta}_{sc}, \hat{\gamma}_{sc}, \hat{t}_{sc}\} = \underset{\alpha, \beta, \gamma, t}{\operatorname{argmax}} \left[ MVN\left( g(\overline{\mathbf{Y}_{sc}^{affected}} - \mathbf{h}_{sc}; \lambda_{sc}); \hat{\boldsymbol{\mu}}_{sc}^{affected}, \hat{\boldsymbol{\Sigma}}_{sc}/n_{affected} \right) \right]$$

where $n_{affected}$ is the number of blink-affected samples, and $\overline{\mathbf{Y}_{sc}^{affected}}$ is the mean of $\mathbf{Y}_{jsc}^{affected}$ over all those samples. Here, the power transform of the subtraction of BPR from the mean of all the blink-affected samples can be approximated as below:

$$g\left( \overline{\mathbf{Y}_{sc}^{affected}} - \mathbf{h}_{sc}; \lambda_{sc} \right) \sim MVN\left( \hat{\boldsymbol{\mu}}_{sc}^{affected}, \hat{\boldsymbol{\Sigma}}_{sc}/n_{affected} \right)$$

Because

$$\frac{1}{n_{affected}}\sum_{j=1}^{n_{affected}} g\big(\mathbf{SF}_{jsc} + \mathbf{PRC}_{jsc}^{affected}; \lambda_{sc}\big) \sim MVN(\hat{\boldsymbol{\mu}}_{sc}^{affected}, \hat{\boldsymbol{\Sigma}}_{sc}/n_{affected}).$$

The inference of $\{\alpha_{sc}, \beta_{sc}, \gamma_{sc}, t_{sc}\}$ allows for replacing $\mathbf{h}_{sc}$ with $\hat{\mathbf{h}}_{sc}$, from which the power transform of the BPR-affected time courses can be derived as follows:

$$g\big(\mathbf{Y}_{jsc}^{affected} - \mathbf{BPR}_{jsc}; \lambda_{sc}\big) = g\big(\mathbf{Y}_{jsc}^{affected} - \boldsymbol{\theta}_{jsc} * \hat{\mathbf{h}}_{sc}; \lambda_{sc}\big) =$$
$$g\big(\mathbf{SF}_{jsc} + \mathbf{PRC}_{jsc}^{affected}; \lambda_{sc}\big) \sim MVN(\hat{\boldsymbol{\mu}}_{sc}^{affected}, \hat{\boldsymbol{\Sigma}}_{sc}).$$

Lastly, $\boldsymbol{\theta}_{jsc}$, the blink-by-blink amplitude of $\mathbf{h}_{sc}$, can be inferred by finding the value of $\boldsymbol{\theta}_{jsc}$ that maximizes the power transform of the BPR-affected time courses:

$$\hat{\boldsymbol{\theta}}_{jsc} = \underset{\theta}{\operatorname{argmax}} \left[ MVN\Big( g\Big(\mathbf{Y}_{jsc}^{affected} - \boldsymbol{\theta}_{jsc} * \hat{\mathbf{h}}_{sc}; \lambda_{sc}\Big); \hat{\boldsymbol{\mu}}_{sc}^{affected}, \hat{\boldsymbol{\Sigma}}_{sc}\Big) \right].$$

We note that when two or more blinks occurred within 3 s, BPRs were overlapped with one another, which made it hard to estimate non-BPR components. Thus, in such cases, $\hat{\mu}_{sc}^{affected}$ was replaced with a vector whose elements were padded with the first or last values at each blink, and then averaged across blinks. We implemented this BPR-correction algorithm into MATLAB scripts (bprcorrect.m in the BPR-correction toolbox) as an out-of-the-box solution so that users can readily apply the algorithm to their data (https://github.com/yookyung1310/BPR_toolbox).

## Supporting information

**S1 Fig. Individual's BPR shape remains constant across runs.** (left panel) Split-half correlation in peak amplitudes (right panel) Split-half correlation in peak time.
(TIFF)

**S2 Fig. Zoomed out diagonal plot for Fig 9B and 9C.**
(TIFF)

**S1 Video. Example of blink-locked pupillary response (BPR).**
(MOV)

## Acknowledgments

This article originated as part of a dissertation completed at Seoul National University by the first author under the direction of the last author. We acknowledge Joonwon Lee and Jaeseob Lim for advising some analyses and visualization and Minjin Choe for part of data collection.

## Author Contributions

**Conceptualization:** Sang-Hun Lee.

**Data curation:** Kyung Yoo.

**Formal analysis:** Kyung Yoo.

**Funding acquisition:** Sang-Hun Lee.

**Investigation:** Kyung Yoo, Jeongyeol Ahn.

**Methodology:** Kyung Yoo.

**Resources:** Sang-Hun Lee.

**Software:** Kyung Yoo.

**Supervision:** Sang-Hun Lee.

**Validation:** Kyung Yoo.

**Visualization:** Kyung Yoo.

**Writing – original draft:** Kyung Yoo, Jeongyeol Ahn, Sang-Hun Lee.

**Writing – review & editing:** Kyung Yoo, Jeongyeol Ahn, Sang-Hun Lee.

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
