## [Decision Letter · Decision Letter 0]

13 May 2021

PONE-D-21-12636

The Confounding Effects of Eye Blinking on Pupillometry, and a Remedy for Them

PLOS ONE

Dear Dr. Lee,

Thank you for submitting your manuscript to PLOS ONE. After careful consideration, we feel that it has merit but does not fully meet PLOS ONE’s publication criteria as it currently stands. Therefore, we invite you to submit a revised version of the manuscript that addresses the points raised during the review process.

Thank you for submitting this article, which has now been reviewed by three reviewers. All three reviewers are generally positive about the work, and make concrete suggestions. All points raised by the reviewers should be addressed comprehensively in your revision.

We look forward to receiving your revised manuscript.

Kind regards,

Manuel Spitschan

Academic Editor

PLOS ONE

Journal Requirements:

3.Thank you for stating the following in your Competing Interests section: 

"No"

Reviewers' comments:

Reviewer's Responses to Questions

**Comments to the Author**

1. Is the manuscript technically sound, and do the data support the conclusions?

Reviewer #1: Yes

Reviewer #2: Partly

Reviewer #3: Yes

2. Has the statistical analysis been performed appropriately and rigorously? 

Reviewer #1: Yes

Reviewer #2: Yes

Reviewer #3: I Don't Know

3. Have the authors made all data underlying the findings in their manuscript fully available?

Reviewer #1: Yes

Reviewer #2: Yes

Reviewer #3: Yes

4. Is the manuscript presented in an intelligible fashion and written in standard English?

Reviewer #1: Yes

Reviewer #2: Yes

Reviewer #3: No

5. Review Comments to the Author

Reviewer #1: # Review

Yoo, Ahn, & Lee (2021) The confounding effects of eye blinking on pupillometry, and a remedy for them. *PLoS ONE*.

---

In this article, the authors describe the blink-induced pupil response (BPR), which is a pronounced pupil constriction from about 200 to 3000 ms after a blink. They point out that this not only increases noise when measuring pupil size, but is also likely to introduce confounds when blink rate covaries with the experimental conditions (as it often does). They then outline a technique to remove the PBR from the pupil time series, and show that this increases statistical power.

I feel that this is a very well-done study. It's well-written, despite some sentence-level English mistakes (but the structure is well done). And the aim and conclusions are clear and reasonable. I have a few suggestions to further strengthen the manuscript.

The introduction does not address the question of *why* the pupil constricts after blinks. In a sense, this is not important, because in the context of the current manuscript it's mostly relevant that it does (and not necessarily why). However, I suspect most readers will still be left wondering, and so I would add a short paragraph about this. I know this probably comes across as a cheap excuse to ask for a citation (I don't mean it as such), but I speculated on this in a recent review article, in which I basically suggest that it's the same phenomenon as the pupil constriction that is induced by visual change:

- <https: 10.1146="" annurev-vision-030320-062352="" doi="" full="" www.annualreviews.org="">

More importantly, I feel that the manuscript would benefit from practical guidelines for researchers to actually apply this method. It seems that the OSF repository includes code to conduct the analyses, or at least there's a `codes.zip` file. But this file seems to be corrupted, and in any case the organization of the OSF and to what extent it contains useful code is unclear. So cleaning up and documenting the OSF project would be low-hanging fruit.

And what kind of practical considerations should researchers keep in mind? For example, when using an EyeLink, you generally start recording at the start of a trial, and then end recording at the end of the trial. This means that there's no continuous pupil time series that spans multiple trials. To what extent is this a problem for applying the proposed technique? More generally, a paragraph with some hands-on advice would be worthwhile, in my opinion.

---

Sebastiaan Mathôt \\

Department of Experiment Psychology \\

University of Groningen \\

<http: smathot="" www.cogsci.nl=""></http:></https:>

Reviewer #2: In the present study the authors explain the BPR problem, the possible artifact that it might cause and also an algorithm to solve it. While the main idea is important and should be discussed and solved, I found three main issues that should be addressed before publication. Please see my comments attached in a PDF file.

Reviewer #3: The Confounding Effects of Eye Blinking on Pupillometry, and a Remedy for Them

The manuscript describes the phenomenon of the blink-locked-pupillary response (BPR), which is the transient pupil constriction that occurs after an eye blink. In cognitive pupillometry research, the BPR is often overlooked completely or regarded simply as a nuisance artefact that must be removed from analysis; but here the authors suggest that the BPR is a 'serious confounder' with the potential to reduce statistical power or to seriously undermine the validity of pupillometry experiments. The manuscript first explores the general shape of the BPR in a simple fixation task and highlights its variability with respect to luminance, individual subjects and trials. With data simulations, it is subsequently demonstrated how the BPR can confound the pupillary signal in the two cognitive tasks (auditory oddball and delayed orientation estimation). Finally, a novel generative model is proposed to account for the BPR and then applied to 'de-confound' the pupil signal of the BPRs influence in the cognitive tasks. When compared to the common standard of using linear interpolation, the model-based correction is shown to substantially improve statistical power.

This is an important contribution to cognitive pupillometry methods, especially given that only one previous study has experimented with a modelling approach to correct for the BPR. The convincing demonstration of improved statistical power would however be more exciting if I felt that the BPR modelling approach–described by the authors as 'The proper way of correcting pupillary measurements for BPR'–would be easy to implement on my own data. The author's refer to a 'BPR correction toolbox' throughout the manuscript, but it remains unclear whether this is actually something that is being offered. What does a researcher have to do to apply this method to their own data? If it is overly complicated, I confess that I would probably stick with linear interpolation and incur the penalty to statistical power.

On the topic of linear interpolation, it may be worth acknowledging that some approaches do attempt to account for the BPR on a blink-by-blink basis. For example, I've previously processed EyeLink data with a toolbox called cili (https://github.com/beOn/cili), which has a function to extend blink end points to the first sample where the z-scored rate of change of the pupil time course drops below 10% of the average within a 100 ms moving window. This does a fairly good job of drawing a line across the BPR and is very easy to implement.

I therefore have the following general recommendations:

1. Describe the steps that a researcher must take in order to implement this approach on their own pupil data, and indicate whether a 'BPR correction toolbox' will be made available. If there is not going to be a toolbox and it is time-consuming to implement, this should be listed as a limitation.

2. Indicate whether there are some situations that this modelling approach would be impractical.

3. Acknowledge alternatives to dealing with the BPR, such as the linear interpolation approach on a blink-by-blink basis that I mentioned above.

Is the manuscript technically sound, and do the data support the conclusions?

The manuscript appears technically sound with data that support the conclusions. I have the following minor suggestions for improvement:

1. With EyeLink systems, optical distortion of the pupil image (caused by changes in gaze position) affects the pupil data by up to 10%. This should be acknowledged, and ideally a simple analysis provided showing that participants performed the fixation task adequately. It would also be helpful to indicate whether gaze related trial exclusion criteria were applied for the fixation and oddball tasks.

2. Please clarify whether any of the participants took part in more than one of the three studies outlined in the manuscript.

3. Please indicate whether a 'BPR toolbox' will be made available, and if so, where it is maintained and and how to use it.

4. Corneal flux density (c) is included as a parameter in the model. Does this mean it is necessary for researchers to obtain their own measurements of corneal flux density in order to accurately model the BPR?

Has the statistical analysis been performed appropriately and rigorously?

The analysis appears to be in good order, but I am not an expert in modelling and therefore do not feel qualified to make comments on the finer details of the analysis. My general feeling is that this may be overcomplicated for the routine analysis of pupillometry data.

Minor points:

1. The meaning of the small lines on the x-axis in Figure 2 should be clarified in the figure caption. I assume they are indicating the time-to-peak for the respective traces?

2. The readability of the figures could be improved by increasing the DPI and possibly font size.

Have the authors made all data underlying the findings in their manuscript fully available?

Yes, all code and data are available on the OSF.

Is the manuscript presented in an intelligible fashion and written in standard English?

The manuscript is presented in an intelligible fashion but lapses frequently in grammar and phrasing, sometimes tone. With this in mind, I recommend additional proof-reading and copy editing. Special attention should be given to the abstract, figure captions and the Materials and Methods Section.

Example minor points:

1. Line 45 - BPR is defined here as blink-locked pupillary response, but was defined as as blink-induced pupillary response in the abstract (line 22).

2. Line 61 - (about xx in z-score unit) - Estimated value is missing

3. Line 563 - 'eyet racker'

4. Line 580 - 'Fixaion'

5. Line 626 - Review this sentence

6. Line 706 - 'we collected acquired'

7. Lines 457-460 - review sentence

8. Line 481 - 'In average'

9. Line 502 - 'vicious confounder' - revise word choice?

10. Line 564 - 'corneal reflex' - shouldn't this be 'corneal reflection'?

6. PLOS authors have the option to publish the peer review history of their article (what does this mean?). If published, this will include your full peer review and any attached files.

Reviewer #1: **Yes: **Sebastiaan Mathôt[Review attached as PDF]

Reviewer #2: **Yes: **Ronen Hershman

Reviewer #3: **Yes: **Joel T. Martin

---

## [Author Response · Author response to Decision Letter 0]

11 Jul 2021

We previously submitted a manuscript titled “The Confounding Effects of Eye Blinking on Pupillometry, and a Remedy for Them” as a Research Article to PLOS ONE.

Our manuscript had been reviewed, and the editor asked us to revise the original manuscript by addressing the reviewers’ comments. We thank the reviewers for their constructive comments on the previously submitted version of our manuscript. 

The co-authors and I carefully studied the reviewer’s comments, and carried out additional analyses on the data and conducted further literature search to fully address the concerns raised in those comments. Based on these new analyses and literature search, we revised our manuscript, including Abstract, Introduction, Results, Discussion, Materials and Methods, with an additional figure and several additional panels in the old figures. We also modified the title of the manuscript (newly titled “The Confounding Effects of Eye Blinking on Pupillometry, and their Remedy”).

Thanks to the reviewers’ comments, we believe that the revised manuscript now presents our findings on more solid methodological grounds and in a form more accessible to the researchers in the field of cognitive pupillometry than its previous version.

Please see the attached file (titled "Responses to Reviewers") for our point-by-point replies to the reviewers' comments.

---

## [Decision Letter · Decision Letter 1]

6 Oct 2021

PONE-D-21-12636R1The Confounding Effects of Eye Blinking on Pupillometry, and their RemedyPLOS ONE

Dear Dr. Lee,

Thank you for submitting your manuscript to PLOS ONE. After careful consideration, we feel that it has merit but does not fully meet PLOS ONE’s publication criteria as it currently stands. Therefore, we invite you to submit a revised version of the manuscript that addresses the points raised during the review process.

Please respond to the queries from Reviewer #2.

We look forward to receiving your revised manuscript.

Kind regards,

Manuel Spitschan

Academic Editor

PLOS ONE

Journal Requirements:

Additional Editor Comments (if provided):

I would kindly ask you to respond to the remaining questions from Reviewer 2.

Reviewers' comments:

Reviewer's Responses to Questions

**Comments to the Author**

1. If the authors have adequately addressed your comments raised in a previous round of review and you feel that this manuscript is now acceptable for publication, you may indicate that here to bypass the “Comments to the Author” section, enter your conflict of interest statement in the “Confidential to Editor” section, and submit your "Accept" recommendation.

Reviewer #1: All comments have been addressed

Reviewer #2: (No Response)

Reviewer #3: All comments have been addressed

2. Is the manuscript technically sound, and do the data support the conclusions?

Reviewer #1: Yes

Reviewer #2: Partly

Reviewer #3: Yes

3. Has the statistical analysis been performed appropriately and rigorously? 

Reviewer #1: Yes

Reviewer #2: No

Reviewer #3: Yes

4. Have the authors made all data underlying the findings in their manuscript fully available?

Reviewer #1: Yes

Reviewer #2: Yes

Reviewer #3: Yes

5. Is the manuscript presented in an intelligible fashion and written in standard English?

Reviewer #1: Yes

Reviewer #2: Yes

Reviewer #3: Yes

6. Review Comments to the Author

Reviewer #1: I feel that the authors have adequately addressed all my comments. This is a very useful manuscript for pupillometry researchers.

Reviewer #2: I have read the authors' responses and the revised manuscript and find that most of my concerns are still there. Moreover, it made me wonder about additional issues that I didn’t notice in the first version. Please see my comments to the authors in the attached PDF.

Reviewer #3: Having read the revised manuscript I feel that all of the comments were addressed and that the paper should be accepted for publication. I was also able to run the MATLAB toolbox tutorial without issue.

7. PLOS authors have the option to publish the peer review history of their article (what does this mean?). If published, this will include your full peer review and any attached files.

Reviewer #1: **Yes: **Sebastiaan Mathôt

Reviewer #2: **Yes: **Ronen Hershman

Reviewer #3: **Yes: **Joel T. Martin

---

## [Author Response · Author response to Decision Letter 1]

29 Nov 2021

General comments ([G1] ~ [G5])

“I have read the authors' responses and the revised manuscript and find that most of my concerns are still there. Moreover, it made me wonder about additional issues that I didn’t notice in the first version. 

As I mentioned in my previous review, while the main idea is important and should be discussed and solved, there are several issues that should be addressed before publication should be considered. 

[G1] The most important issue is the validity of the algorithm. Specifically, the authors presented partial analysis (i.e., only for two experiments) that estimated the efficiency of their algorithm. [G2] Another important issue is about the way they estimated the efficiency of their algorithm; namely, what they compared (or didn’t compare). In my opinion, there are difficulties that make the conclusion unclear. [G3] The third main issue (and maybe the most serious one) is about the methodology that the authors used in their studies. Specifically, it seems that the second experiment had short durations that cannot be used in pupillometry studies (due to lack of time to get back to baseline). [G4] In addition, it seems there are several methodological issues in their analysis; specifically, the smoothing that might destroy valid data, comparison that cannot provide information about the influence of the manipulation on the pupil size, and possibly wrong blink corrections. These are all a little bit problematic in a study that aims to solve a methodological issue. [G5] Another important issue is that the authors didn’t provide any real indication of the importance of the BPR correction. Yes, indeed, statistical power is highly important, but without a clear-cut example for the problematic uncorrected BPR, I can’t see any reason to adopt the authors’ algorithm. In other words, if it is impossible to present an example that will show how the correction helps researchers, there is no actual reason to use it.”

Our replies to the general comments ([G1]~[G5]): As indicated right below, the reviewer here gives a brief and general summary of the key issues that will be specified in the following specific comments. Specifically, [G1] and [G2] appear to be specified mainly in his comment labeled with [C2-1]; [G4] in [New15]; [G5] in [C2-2]. Thus, our replies to [G1]~[G5] will be addressed in our point-by-point replies to those specific comments. 

Although we appreciate (and agree with) a few of the “comments” newly made in the second round of revision and revised the manuscript accordingly, we, in general, do not agree with the reviewer’s main assertions summarized in [G1]~[G5]. The corresponding grounds for our disagreement with [G1]~[G5] will be presented in our replies to the reviewer’s specific comments, which consist of the reviewer’s rebuttals to our previous replies to the reviewer’s comments in the first round of review (a total of 5 comments, [C2-1], [C2-2], [M2-3], [M2-4], and [M2-6]) and the reviewer’s new comments in the second round of review (a total of 5 comments, [New1]~[New17]).

However, [G3], the comment about the second (auditory oddball detection) experiment was not specified in any of the following specific comments. Thus, here we provide our reply to [G3].

[G3]: “The third main issue (and maybe the most serious one) is about the methodology that the authors used in their studies. Specifically, it seems that the second experiment had short durations that cannot be used in pupillometry studies (due to lack of time to get back to baseline).”

Our reply to [G3]: We do not understand why the second experiment threatens the validity of our work. We would like to remind the reviewer of the purposes of the second experiment, as written in the current manuscript: (1) “To specifically illustrate how BPR can confound the pupillary signal of interest, we measured the pupil size while subjects were performing two cognitive tasks, ‘auditory oddball detection’ and ‘delayed orientation estimation’ tasks.” (the second paragraph of the Results section titled “BPR, not just a nuisance but a confounder”); (2) “As the second step of verifying the effectiveness of the correction algorithm, we applied it to the data acquired in the auditory oddball task.” (the first paragraph of the Results section titled “Correcting the auditory oddball data for BPR”. As explicitly stated in (1), the first purpose of the auditory oddball detection experiment was to concretely show an example case in which spurious increments or decrements appear depending on the timing of BPR when researchers design experiments with a short inter-trial interval and adopt the practice of adapting the pupillary time course to the pupil size at trial onset. Importantly, this design and practice have been exercised in previous studies, including those cited in the current manuscript. For example, Hong and his colleagues (Hong L, Walz JM, Sajda P (2014) Your Eyes Give You Away: Prestimulus Changes in Pupil Diameter Correlate with Poststimulus Task-Related EEG Dynamics. PLoS ONE 9(3): e91321.) measured the pupil size with an auditory oddball task in which ITI was 2 or 3 seconds and quantified the evoked pupil responses by adapting the trial-chopped time course of pupil size to the trial onset (as seen in their Figure 3). There have been several pupillometry studies like Hong et al. (2014), such as Keute et al. (2019) (Keute, M., Demirezen, M., Graf, A., Mueller, N. G., & Zaehle, T. (2019). No modulation of pupil size and event-related pupil response by transcutaneous auricular vagus nerve stimulation (taVNS). Scientific reports, 9(1), 1-10.) and Nakakoga et al. (2021) (Nakakoga, S., Shimizu, K., Muramatsu, J., Kitagawa, T., Nakauchi, S., & Minami, T. (2021). Pupillary response reflects attentional modulation to sound after emotional arousal. Scientific reports, 11(1), 1-10.).The second purpose of the auditory oddball detection experiment was to demonstrate the effectiveness of our BPR correction algorithm by showing that the spurious decrements and increments due to BPR almost disappeared when the BPR correction algorithm was applied. Thus, the data and analysis outcomes of the auditory oddball detection experiment supported the claims we make in the current study. Then, why are the data and the analysis carried out on the auditory oddball experiment “the most serious” problems?

Specific comments originating from the first round of reviews ([C2-1], [C2-2], [M2-3], [M2-4], and [M2-6])

 As I mentioned above, I read the author's comments and the revised manuscript and would now like to refer to specific difficulties that I found in their responses and in the manuscript. First, I would like to address the authors’ responses to the main comments (the identification of the comment is what the authors used in their response).

 [The issues related to the major point [C2-1] in the first-round of review/revision]

(To clarify the origin of issues and track the past exchanges between the reviewer and us, we copied and pasted the corresponding parts from our first-round point-by-point replies.) 

The reviewer’s original comment [C2-1]: “In the present study the authors explain the BPR problem, the possible artifact that it might cause and also an algorithm to solve it. While the main idea is important and should be discussed and solved, I found three main issues that should be addressed before publication.

1. The authors present and analyze the pupil size that was changed due to the BPR. However, it’s

not clear whether eye blinks were caused / associated with the observed pattern. For example, in

Fig. 5C it seems that there are dips after about 5 seconds and 20 seconds. These dips looks pretty

much the same as other dips that were associated by the authors to eye blinks and therefore to

BPR. Moreover, it seems that there are oscillations in pupil size that cause dipping about every 5

seconds (0.2 Hz). There are several possible options to solve this problematic issue. One possible

options is to compare two groups – one group that will be allowed to blink, and another group

that won’t be able to blink (by using dedicated eye drops). Another possible option is by

comparison between trials / time windows with blinks and those without blinks. If the authors

find meaningful differences between the groups, the interpretation of the original results will be

more reliable. Without this comparison it is difficult to understand the weight of this possible

artifact in pupillometry studies.”

Our reply to [C2-1] in the first-round revision: We agree with the reviewer that the weight of BPR in pupillometry would be better appreciated if there is a more straightforward demonstration of the effects of BPR correction. We thank the reviewer for suggesting two possible options for making such a demonstration. We reasoned that the second option, i.e., comparing the ‘blink-affected’ trials and the ‘blink-free’ trials, is more suitable than the first option, i.e., comparing the ‘blink-allowed’ condition and the ‘blink-suppression’ condition. Of course, the first option requires additional data collection. However, a more important reason for preferring the second option to the first one is our concern that active suppression of blinks is likely to distort the genuine time course of pupil size (see a new section of DISCUSSION in the revised manuscript (“Practical considerations for avoiding BPR-related problems”, which was added as our reply to one of the Reviewer #1’s comments ([C1-4])). So, we carried out the second analysis on the auditory oddball detection data (wherein individual trials are brief (2 s) and thus ideal for trial-to-trial analysis) and summarized the results with a new paragraph (the fourth paragraph under the RESULTS section titled “Correcting the auditory oddball data for BPR”) and a new figure (Figure 6) in the revised manuscript.

 We note that the reviewer, while raising a possibility that the results shown in Figure 5 might not indicate the association (or causal relationship) between blinks and BPR, pointed out the wavy modulation of pupil size in the BPR-corrected time course of pupil size (“For example, in

Fig. 5C it seems that there are dips after about 5 seconds and 20 seconds. These dips looks pretty

much the same as other dips that were associated by the authors to eye blinks and therefore to

BPR. Moreover, it seems that there are oscillations in pupil size that cause dipping about every 5

seconds (0.2 Hz).”). This wavy time course was referred to again in one of the minor points of the reviewer (see [M2-5] in the below: “5. Lines 603-609: Why were the fixation colors different between trials? With respect to the first major point, maybe these contrasts were the reason for the 0.2Hz oscillations.”). We were quite perplexed with this comment and cautiously guess that the reviewer was confused about the time scale at which the time course of pupil size is described: the time unit used in all the plots of Figure 5 was “second” and thus the wavy modulation occurs not at 0.2 Hz but at 2 Hz. If we misunderstood the reviewer’s point, please let us know. However, more importantly, the reviewer’s pointing out this wavy modulation (and associated possible suspicion that comes with it) made us think that it’s better to offer some explanation for the wavy curve (we thank the reviewer for that). Although not 100% sure, we think that the first and second peaks are likely to reflect the previously known pupil responses that are associated with a sound stimulus and a manual response to it, given that subjects heard a sound and judged the tone of the sound by pressing one of the two assigned keys each and every trial. This conjecture is in line with the results that the blink-free trials (the gray lines in Figure 6A in the revised manuscript) and the corrected blink-affected trials (the green line in Figure 6A) both exhibited the double-peak pattern at 2 Hz. And we looked up the literature on pupillometry experiments with the auditory oddball task and found that the wavy pattern has been reported in the studies where subjects pressed a button on each trial like in our study [1,2]. In the revised manuscript, we offered our interpretation for this wavy modulation at 2 Hz in the fourth paragraph of the RESULTS section titled “Correcting the auditory oddball data for BPR”.

The reviewer’s rebuttal to our reply to [C2-1] in the first round of revision: “[C1-2](probably mislabeling [C2-1] with [C1-2]): I don’t really understand the rationale behind the authors’ test. In each trial there were two options: there was a correction of BPR (by using the authors’ algorithm) or there was no correction. Why did the authors choose a different separation of trials? Moreover, in my opinion the analysis should be applied to all the experiments (to support the existence of the problem) and not only to the first two experiments. In addition, the comparisons have to be identical across experiments in terms of separation criteria (related to [G1][G2]). Actually, the presented analysis (see more comments about the presented analysis below) does not answer the important question: Does the authors’ algorithm solve the problem?

I think I wasn’t clear in my previous review so I’ll try to explain it now. In Figure 4B (the one with a time window of 35 seconds), there is a decrement of the pupils about every 5 seconds = 0.2 Hz. Hz means occurrences per second. 2Hz means 2Hz occurrences per second. Here there is less than one occurrence per second. Hence, the frequency has to be smaller than 1. In Figure 4B there is one occurrence every 5 seconds, namely 1/5s = 0.2Hz. These decrements (or oscillations in the terminology of the original comment) are very strange and made me think maybe it might explain part of the results of the authors. These decrements (as far as reflected in the figure) are not associated with eye blinks (because their number is larger than the number of the observed eye blinks). If this is the case, I’m not entirely sure what the meaning of the analysis is and moreover, what the validity of the present study is.”

Our re-reply to the reviewer’s rebuttal to our reply to [C2-1] in the first round of revision (Although the reviewer labeled this rebuttal with [C1-2], he seemed to mean [C2-1] by it when the context was considered and because [C1-2] refers to the second major comment of the other reviewer (Reviewer 1).):

Regarding the issue of “rationale of separation of trials”. The reviewer repeatedly stated that it is difficult to understand the rationale behind our comparing the BPR-affected time course of pupil size against that of the BPR-free time course of pupil size before and after correction, respectively. We have a clear rationale for this comparison, and there is a reason why we thought this comparison is appropriate for the fixation and the auditory oddball experiments but not so for the delayed orientation estimation experiment, as follows.

One important merit of the data set acquired from the first two experiments (i.e., fixation and auditory oddball detection) is the fact that an effective ground-truth time course of pupillary responses, which are free from BPR, can be approximated by the average of the sufficiently long periods during which blinks did not occur (see Materials and Methods for the detailed description of BPR-free time courses). It is important to compare the BPR-affected time course of pupillary responses against this ground-truth time course before and after, respectively, the correction for BPR because it allows us to verify whether our correction algorithm selectively removes the pupillary responses associated with BPR but not those associated with other components, including spontaneous fluctuations. In this regard, note that the blink-free, ground-truth time course of pupillary responses might not be necessarily flat because there could be many uncontrolled factors that potentially affect the pupillary responses (e.g., gradual decrement or increment in pupil size over time due to fatigue or arousal, or peculiar fluctuations associated with task structure). In such cases, the simple comparison of the pupillary responses before and after the application of BPR correction (, which seems to be suggested by the reviewer based on his comment, “In each trial there were two options: there was a correction of BPR (by using the authors’ algorithm) or there was no correction. Why did the authors choose a different separation of trials?”) does not provide the sufficient information about whether the algorithm successfully corrected the pupillary responses only for BPR, which is the exact goal of the BPR correction algorithm. In the revised manuscript, we added several sentences by explicitly stating why this comparison is required both for the fixation experiment (the last paragraph of the Results section titled “Model-based correction of pupillary responses for BPR”) and for the auditory oddball detection experiment (the fourth paragraph of the Results section titled “Correcting the auditory oddball data for BPR”).

The reviewer also asked why we didn’t apply this comparison for the delayed orientation estimation task. There were two main reasons. First, the trial structure was not suitable for this type of comparison. Unlike the other two experiments, a single trial lasted for more than 10 seconds (including the phase of confidence reporting and inter-trial interval; see Methods & Materials), which made it virtually impossible to find blink-free trial because blink occurs typically several times during single trials. Second, the main purpose of the delayed orientation estimation experiment was to demonstrate (i) a case in which BPR can directly confound the independent variables of interest and (ii) our algorithm can be effective in de-confounding the pupillometry data by selectively filtering out BPR component. That’s why we focused on comparing the differences between the different levels of the independent variables before and after BPR correction. As mentioned below, what’s important is the fact that we used the same algorithm consistently and successfully demonstrated that our algorithm can be applied for BPR correction in two example experimental situations where our algorithm helps experimenters to get rid of spurious increments and decrements in pupil size (in the auditory oddball detection experiment) and de-confound the confounded pupillometry signals (in the delayed orientation estimation experiment).

Regarding the issue of “applying the same analysis to the three experiments”. We do not agree with the reviewer’s assertion that trials should be separated exactly in the same way for all the experiments. We neither agree that the same analysis should be applied to all the experiments. What’s important is - not to apply the one and only analysis repeatedly to different data sets - but to choose a type of analysis that is appropriate for a given purpose.

To be sure, we applied the same algorithm to the three sets of data (namely, the fixation, auditory oddball, and delayed orientation estimation task) but wanted to demonstrate, firstly, the diverse contexts in which BPR occurs and acts as confounders and, secondly, that our proposed algorithm is effective in correcting the pupil time courses for BPR under those different contexts. The three data sets were acquired with different purposes and can be examined using different analyses or comparisons that best serve their respective purposes. We do not believe there is such a rule dictating that one and only type of analysis should be applied to the data sets belonging to a given study.

In the fixation experiment, there was no cognitive task except for constant (i.e., time-invariant) fixation, which creates an ideal situation where the effective ground-truth time course that is affected neither by BPR nor by task-related responses can be acquired and utilized as a reference for verifying the effectiveness of the algorithm. That’s the reason why we opted to compare the BPR-corrected time course to the effective ground-truth time course (as described in the last paragraph of the Results section titled “Model-based correction of pupillary responses for BPR” and Fig 4C,D). Unlike in the fixation experiment, subjects performed specific cognitive tasks in the remaining two experiments, namely the auditory oddball detection and delayed orientation estimation tasks. The two experiments create quite different contexts, especially in trial length. Because the trial length is shorter than the length of BPR (>3 sec) in the auditory oddball experiment, we predicted, via simulation based on our generative model (Fig 4A,B), that the practice of adapting the trial-chopped pupil size time courses to the pupil size at trial onset would generate spurious upward or downward fluctuations due to BPR depending on when a blink occurs unless the raw time courses are corrected for BPR. That’s why we separated four consecutive trials into two different sequences ([0 0 1 0] and [0 0 0 1]) and tested whether the proposed algorithm can get rid of such spurious fluctuations (as described in Fig 4C). Note that this type of analysis is neither appropriate nor possible in the fixation or delayed orientation estimation experiments because there was no trial structure in the former and the trial length was much longer than the length of BPR in the latter (as mentioned above). Next, in the delayed orientation estimation experiment, we focused on a context where the blink rate is associated with the main cognitive variable, i.e., working memory load – the number of memoranda to keep in mind. Here, we focused on the task phase in which such an association was apparent (right after estimation onset, when the confounding due to BPR are expected to occur, as shown in the top panel of Fig 3D and Fig 9A), predicted, via simulation based on the generative model (Fig 4C,D), that the association between blink rate and memory load would underestimate the changes in pupil size between the different memory load conditions, and tested whether the proposed algorithm can address such underestimation due to BPR (as described in Fig 9). 

Put together, we demonstrated the effectiveness of the proposed algorithm in different contexts by applying the analyses that were tailored to those contexts. Contrary to what the reviewer asserted, the approach taken by the current work has a merit, rather than a problem, by showing that it can be applied to different situations to address various problems due to BPR. We could not find a reason to follow reviewer’s strict request that “the [same] analysis should be applied to all the experiments”. It is “an algorithm” but not “an analysis” that must be applied consistently.

Regarding the issue of “0.2 Hz fluctuations”. On a separate note, we should stress that the reviewer initially (on the first round of review) referred to Fig 5C, not to Fig 4B, when describing 0.2 Hz fluctuations (look above (The reviewer’s original comment [C2-1]) for those highlighted in red). In our previous reply (Our reply to [C2-1] in the first-round revision), we couldn’t find 0.2 Hz fluctuations in Fig 5C and instead guessed that the reviewer might have referred to 2Hz ripple-like fluctuations. From our perspective, this issue comes as completely different issues. We clarify that this is not our fault but the reviewer’s fault, which shouldn’t be tersely excused with a comment like “I think I wasn’t clear in my previous review.” We spent a lot of time due to the reviewer’s mistake, being puzzled to figure out what the reviewer means by that. Unfortunately, we now have to deal with another totally different issue raised by the reviewer (, again from our standpoint to be sure).

Anyway, we now understand what the reviewer implies regarding Fig 4B. However, we do not agree with the reviewer’s assertion that the presence of decrements that are not associated with blinks threaten the validity of our work. In the current work, we never claimed that our algorithm can fix all the problematic fluctuations existing in pupillometry data (e.g., fluctuations associated with saccadic or micro-saccadic eye movements). Instead, what we do claim in the current work is that (i) blinks are followed by a short-pipe-shape fluctuation of pupil size (BPR), which varies in amplitude across blinks and in shape across individuals; (ii) BPR is not just a nuisance but a confounder because blinks do not randomly occur but can be associated with task phases or experimental conditions; (iii) the proposed model-based algorithm can effectively (we never said ‘perfectly’ or ‘efficiently’, to be sure) remove BPR without affecting non-BPR components in pupillary responses. We stated clearly throughout the entire manuscript that these three are our main claims, i.e., “the meaning of the analysis”. The decrements that are not associated with blinks might be due to (micro)saccades or spontaneous pupil fluctuations (we do not know). However, their presence and addressing them are simply beyond the scope of the current study. Thus, we do not find a reason why the reviewer thinks the presence of the decrements, which are based on the reviewer’s anecdotal and subjective eyeballing observation, threatens “the validity of the present study”, and why the current study should address them. 

By the way, we confirmed that the 0.2Hz – seemingly oscillatory – fluctuations, which the reviewer generalized from the sample time course of pupil size depicted in Fig. 4B, did not exist in the data collected in the fixation experiment: our power spectrum analysis did not indicate any peculiar increase in power at around 0.2 Hz (see the figure in the right). Line and shade indicate the mean and standard errors of the mean (SEM) across runs.

[The issues related to the major point [C2-2] in the first-round of review/revision]

The reviewer’s original comment [C2-2]:“2. In addition to the (relatively) steady state measurement of the pupils (the fixation task), the authors ran more two independent cognitive tasks. While the results of the fixation task were relatively replicated in these tasks, the main idea behind the possible artifact is still not clear. Yes, indeed, the mean was pretty much the same and the scattering was decreased (this makes sense with respect to the first main issue that caused interpolation of a lot of dips). However, the influence of this possible artifact on the actual effects unclear. From the study perspective, the BPR might cause systematic pupil decrement that is not associated with the task (this is a bit of a problematic statement due to the fact that the authors cited studies that associated eye blink rate with task difficulty), therefore, effects should appear (false positive) / disappear (false negative) after the BPR correction. If this is the case here, the authors should present these possible scenarios with / without BPR corrections. Specifically, on one hand, the authors should show effects (in terms of the actual task – e.g., easy vs. difficult) that will appear after correction, and on the other hand, the authors should show effects that will disappear after correction. Without these comparisons, the contribution of the correction is not clear (at least in cognitive tasks).”

Our reply to [C2-2] in the first-round revision: As the reviewer pointed out, it would have been a much more straightforward (or more dramatic) demonstration of the effects of BPR correction if we had shown that the effects of interest (i.e., ‘oddity’ and ‘memory load’ in the first and second cognitive experiments, respectively) “appear (false positive) / disappear (false negative) after the BPR correction”. However, the effects were already statistically significant even before the BPR correction in both of the cognitive experiments. That’s also consistent with what has been previously reported. That’s why we decided to focus on the power analysis and identifying the exact contribution of the BPR correction, which was to decrease the variabilities of pupil-size measurements at the subject-to-subject and blink-to-blink levels (as summarized in Figure 6 and 8 in the previously submitted manuscript). We stress that this specific contribution is highly consistent with the results in the fixation experiment, where we found that BPR substantively varied in both shape and amplitude across individuals and mainly in amplitude across blinks within given individuals. We also stress that what were depicted in Figure 6D and 8E in the old manuscript partly demonstrate what the reviewer wanted to see. That is, there is a certain range of trial numbers (or subject numbers in a realistic situation) wherein the effects of interest disappear (insignificant) before correction and appear (significant) after correction. This was one of the motivations for drawing those plots.

Having explained why we focused on the power analysis, we also understand why the reviewer requested a straightforward demonstration of “false positive” or “false negative”. Although we cannot carry out such a demonstration for the reason pointed out above, we thought that it would help readers ‘tangibly sense’ the contribution of the BPR correction if we visually summarize how much the BPR correction increases the effect size in the two cognitive experiments. Thus, in the revised manuscript, we added new figure panels wherein the mean trajectories of pupil size are shown for each level of the main variable and the time courses of t statistics are shown in parallel for the differences between the levels (Figure 7A and 9A) and added several sentences for describing the results (the fifth paragraph of the RESULTS section titled “Correcting the auditory oddball data for BPR” and the second paragraph of the RESULTS section titled “Correcting the delayed orientation estimation data for BPR”).

The reviewer’s rebuttal to our reply to [C2-2] in the first round of revision: “[C2-2]: If I understand correctly, the authors said that the BPR might not cause a statistical problem (in terms of type I / type 2 errors) but could cause a decrease the statistical power. Moreover, in a series of three experiments, they failed to show why the correction (that it is not clear to me whether it is valid or not) is good and important. Yes, indeed, statistical power is important and is the basis of any statistical test. However, in the presented work, I can’t see any reason to use it. This is because the problem (i.e., in what scenarios the BPR will cause trouble) is not clear or well defined, and because the correction wasn’t compared (convincingly) with different valid tools / cases without BPR correction.” (related to [G5])

Our re-reply to the reviewer’s rebuttal to our reply to [C2-2] in the first round of revision: We do not agree with the reviewer’s assertion that we “failed to show why the correction (that it is not clear to me whether it is valid or not) is good and important.” As we stated in re-replying to the reviewer’s rebuttal to our reply to [C2-1] in the above, we clearly demonstrated that our algorithm is effective in various contexts: (i) by comparing ‘the BPR-corrected time courses of pupil size’ with ‘the effective ground-truth time courses of pupil size’ (fixation experiment, Fig 4C,D); (ii) by demonstrating that the proposed algorithm can effectively remove the spurious fluctuations that arise from the customary procedure of adapting trial-chunked pupil time series to trial onset (auditory oddball experiment, Fig 5-7), and (iii) by demonstrating that the proposed algorithm can increase the differences between the conditions of interest by de-confounding the raw pupil-size responses from BPR (Fig 9). Based on these results, we’ve shown that our method of correction is not only important in that it deals with BPR that is prevalent and can potentially be a serious confounding factor in cognitive pupillometry experiments, but also good in that it effectively corrects the raw pupil size data for BPR under various contexts. Therefore, we do not understand on what basis the reviewer asserts that we “failed to show why the correction is good and important.”

 Next, contrary to what the reviewer insinuated, the statistical power is intimately associated with type I (false positive, its probability being quantified by alpha) and II (false negative, its fraction being quantified by beta) errors because it becomes increasingly difficult to distinguish between null and alternative hypotheses as the statistical power decreases (Krzywinski, M., Altman, N. Power and sample size. Nat Methods 10, 1139–1140 (2013). https://doi.org/10.1038/nmeth.2738). Considering that cognitive neuroscience literature is infamous for a low level of statistical power (~0.2; Button, K., Ioannidis, J., Mokrysz, C. et al. Power failure: why small sample size undermines the reliability of neuroscience. Nat Rev Neurosci 14, 365–376 (2013). https://doi.org/10.1038/nrn3475), the issue of increasing statistical power is not just important in itself but also critically associated with type I and II errors. In this sense, the reviewer mischaracterizes our reply by stating “the authors said that the BPR might not cause a statistical problem (in terms of type I / type 2 errors) but could cause a decrease the statistical power” (, which is an impossible statement by itself for the reason mentioned above). We said such a statement neither in our reply nor in the manuscript. Actually, as we clearly stated in our reply, the main motivation of carrying out the simulation analysis in which we plotted how statistical power increases as the number of trials increases (Fig7E & Fig9E) was exactly to show, in a principled way, that our BPR correction method can substantially affect both type I and type II errors by increasing the statistical power. The statistical power matters and is often expensive to buy for empirical scientists.

[The issues related to the minor point [M2-3] in the first-round of review/revision]

The reviewer’s original comment [M2-3]: “3. Something doesn’t make sense in Fig. 7. After 4 seconds there are a lot blinks (the EBR is relatively large), but the pupil size is still increased (with / without corrections). How it is in line with the main idea of BPR? Why the pupil is sometimes increased and sometimes decreased?”

Our reply to [M2-3] in the first-round revision: First of all, it should be stressed that the pupil-size measurements are not just an outcome of BPR but also influenced by cognitive and spontaneous states, as we assumed in our generative model (as depicted in Figure 4A). We interpret that the pupil size increases after 4 seconds despite of the increase of EBR because the pupil size tends to increase when subjects are required to (‘cognitively’) retrieve the memory of the target orientation and make manual adjustments for estimation report. Both of these processes are known to increase the pupil size (Hong et al., 2014; Murphy et al., 2011; Robison and Unsworth, 2019). Of course, blinks will decrease the pupil size, but blinks do not occur not that often (~0.3Hz) and overshadowed by the rapid increase due to the cognitive demand. As demonstrated in Figure 9A, when BPR is corrected with our algorithm, the corrected time course was greater than the uncorrected time course. In the revised manuscript, we explained the rapid rise during the estimation epoch of the delayed orientation estimation task (the second paragraph of “Correcting the delayed orientation estimation data for BPR”).

The reviewer’s rebuttal to our reply to [M2-3] in the first round of revision: “The authors’ response is a bit problematic. If there are no decrements after any blink, why do the authors associate the decrements after blinks to blinks – this is the main idea of the study (unless I missed something). If the aim is to define the BPR (in terms of shape and latency), this issue should be of concern. Otherwise, it means that in several cases (that cannot be predicted – at least by me) the algorithm might destroy valid data.”

Our re-reply to the reviewer’s rebuttal to our reply to [M2-3] in the first round of revision:

We do not agree with the reviewer’s assertion, “If there are no decrements after any blink, why do the authors associate the decrements after blinks to blinks – this is the main idea of the study (unless I missed something).” As we clearly stated in our first-round reply to [M2-3], “it should be stressed that the pupil-size measurements are not just an outcome of BPR but also influenced by cognitive and spontaneous states, as we assumed in our generative model (as depicted in Fig 4A),” which is why cognitive scientists acquire pupil size measurements and read out such states from those measurements. Thus, there could be no visible decrements in the across-trial-averaged time course of pupil size if there is a counteracting increase in pupil size that is induced by cognitive factors. Again, as clearly stated in our previous reply, this is a very plausible scenario given that memory retrieval and estimation report “are known to increase the pupil size” and that “blinks do not occur that often (~0.3Hz) and overshadowed by the rapid increase due to the cognitive demand.” Thus, the fact that “there are no decrements after any blink” in the across-trial-averaged time course of pupil size does not necessarily negate the presence of BPR. We already revised the manuscript in the first round of revision to clarify this issue by explaining “the rapid rise during the estimation epoch of the delayed orientation estimation task (the second paragraph of “Correcting the delayed orientation estimation data for BPR”)”.

[The issues related to the minor point [M2-4] in the first-round of review/revision]

The reviewer’s original comment [M2-4]: “4. Lines 569-574: Was the exclusion (of both trials and subjects) also based on task performance? If not, why?”

Our reply to [M2-4] in the first-round revision: We did not exclude trials or subjects based on task performance. Task performance was generally good for all the participants, and we could not find any particular reasons to exclude based on task performance. Instead, we excluded the entire data sets from a few subjects when the recording quality was poor, when subjects dozed off during experiments, or when the blink rate was unusually high. We revised the “Subjects” section of METHODS and MATERIALS and stated that how many subjects are discarded from further data analysis on what basis.

The reviewer’s rebuttal to our reply to one of the minor points [M2-4]: “Performances should be added to support the authors’ claim.”

Our re-reply to the reviewer’s rebuttal to our reply to [M2-4] in the first round of revision: Yes, we provided the data that support our claim at the end of “Subjects” subsection of METHODS and MATERIALS in the revised manuscript. In both tasks, subjects in our study showed a reasonable range of performance in accuracy and RT, which were quite comparable to those reported in the previous work the same tasks. In the revised manuscript, we reported the mean±SD and range [min, max] both in accuracy and RT, with the previous work as references. 

[The issues related to the minor point [M2-6] in the first-round of review/revision]

The reviewer’s original comment [M2-6]: “6. Line 653: How did the authors separate between eye-blinks and other artefacts (e.g., noise / head movements).”

Our reply to [M2-6] in the first-round revision: First of all, as we stated in the “Stimuli and procedure” section of METHODS and MATERIALS, we used a chinrest to avoid head movements and to keep the distance between the eyes and the eye-tracker. Thus, the outputs from the eye-tracker were not that noisy and blinks were readily discriminated from other miscellaneous background noises.

The reviewer’s rebuttal to our reply to one of the minor points [M2-6]: “Every device has noise. It means that any device might show extreme pupillometry values by mistake. My recommendation is to exclude extreme values by using Z-scores (in most cases 2.5 Z-scores) based on the mean and the SD of each trial separately. Please note that eye-movements can also cause missing values that might be associated accidently to eye-blinks (e.g., if the participant looks at the floor). In addition, if the participant touches his/her face (it might happen sometimes), missing values will be observed. Other scenarios that I can think about are associated with moving the head back. Yes, indeed, the chinrest is supposed to decrease the possibility of head movements, but it might happen. I highly recommend the authors check this, specifically in methodological work like this.”

Our re-reply to the reviewer’s rebuttal to our reply to [M2-6] in the first round of revision: We appreciate the reviewer’s thoughtful suggestion of excluding “extreme values by using Z-scores (in most cases 2.5 Z-scores) based on the mean and the SD of each trial separately.” However, we opted not to incorporate this suggestion for the following reasons. First, in our previously published work (Choe KW, Blake R, Lee SH. Pupil size dynamics during fixation impact the accuracy and precision of video-based gaze estimation. Vision Res. 2016 Jan; 118:48-59; Choe KW, Blake R, Lee SH. Dissociation between neural signatures of stimulus and choice in population activity of human V1 during perceptual decision-making. J Neurosci. 2014 Feb 12;34(7):2725-43), we learned that making head motions is extremely rare in our setup with a chin-rest. We did not observe that subjects were off the chin rest during the experiments. If they did, our gaze data would pick up that huge missing data. Additionally, we monitored and recorded subject’s gaze data throughout the entire experiments, but we haven’t observed any large-size shifts in gaze data that are expected by the head or eye or face-touching motions mentioned by the reviewer. Second, even if such large gaze shifts occurred during our experiments, our preprocessing procedure of band-pass filtering would catch such extreme values of changes and filter them out. Finally, and most importantly, we avoided the method of excluding data based on Z-scoring because we learned from the fixation experiment that the observed range of BPR amplitudes go beyond 1 mm, which will readily fall outside -2.5 in z score thus will be discarded. Thus, the Z-scoring procedure is likely to exclude the informative part of BPR-related data. 

Specific comments originating from the second round of review ([New1]~[New17])

Beside my comments to the authors’ answers, I have a (relatively long) list of specific comments (the numbers represent the relevant line in the manuscript).” 

[New1]: “30-38: This pupil-size dynamics, which is dissociated with light”: this is not absolutely true. Steinhauer found an interaction between them (Steinhauer, S. R., Siegle, G. J., Condray, R., & Pless, M. (2004). Sympathetic and parasympathetic innervation of pupillary dilation during sustained processing. International Journal of Psychophysiology, 52(1), 77-86).”

Our reply to [New1]: We thank the reviewer for directing us to this important reference. In the revised manuscript, we removed “which is dissociated with light” (the second sentence of Introduction).

[New2]: “152-155: I looked for studies about the “delayed orientation estimation” and pupillometry, and I didn’t find anything relevant. Moreover, the citation (44) is not relevant to this task. Hence, I’m not entirely sure why this task was chosen nor why the authors declared that “These two tasks were chosen because the cognitive processes underlying task performance are relatively well established, and those processes are known to be tightly associated with pupillary responses.”

Our reply to [New2]: First of all, we used the auditory oddball task and the delayed orientation estimation task because the cognitive processes underlying these tasks, such as ‘surprise by oddity’ and ’ working memory load’, are known to be tightly linked to pupil size. As pointed out by the reviewer, the references 44 and 45 are relevant only to the (auditory) oddball task. In the revised manuscript, we added old and new references (Kahneman D, Beatty J. Pupil Diameter and Load on Memory. Science. 1966;154: 1583–&. doi:10.1126/science.154.3756.1583; Bays PM. Noise in Neural Populations Accounts for Errors in Working Memory. Journal of Neuroscience. 2014;34: 3632–3645. doi:10.1523/JNEUROSCI.3204-13.2014; Denison, R.N., Parker, J.A. & Carrasco, M. Modeling pupil responses to rapid sequential events. Behav Res 52, 1991–2007 (2020)) as references associated with the delayed orientation estimation task, and modified the sentence a little.

[New3]: “156-173: I took a look at the cited paper and the authors in that paper made a different analysis (depending on the luminance). It is difficult to understand whether the results of the authors of the present manuscript are good or not in terms of replication.”

Our reply to [New3]: We are quite puzzled by this comment because the cited references (19, 29-31) were clearly in line with our results by reporting that the blink rate tends to decrease during stimulus presentation or during important task phases and increase during implicit break points. Specifically put, Siegle et al. (2008) showed (in their Figure 1a) that the blink rate decreased during the stimulation phase and then started to increase during the mask period, which can be considered as an implicit break point. Orchard et al. (1991) showed that the blink rate decreased during a cognitively important period (i.e., when fixation pauses during normal reading) and increased during the moment of line change, which can also be considered as an implicit break point. Nakano (2009) showed that the blink rate decreased when subjects viewed important movie scenes but increased when they viewed unimportant movie scenes, as the authors explicitly stated “we found the synchronization of blink timing within and across individuals at implicit breaks in video stories.” Lastly, Liao et al. (2016) showed (in their Figure 7) that the blink rate decreased during stimulus presentation and then increased afterwards. Having indicated the relevance of these references, we slightly modified the sentences considering that Orchard et al. (1991) and Nakano (2009) did not directly show the time course of blink rate as in our study (the last paragraph of the Results section titled “BPR, not just a nuisance but a confounder”). 

 We do not understand the part about luminance (“the authors in that paper made a different analysis (depending on the luminance)”) because we couldn’t find any indication that the authors in the referenced papers carried out different analyses depending on the luminance.

[New4]: “193-204: Again, it is difficult to understand if the results are good or not due to the fact that they weren't compared with previous results in terms of replication.”

Our reply to [New4]: It is true that, to our best knowledge, there is no previous study directly showing that the blink rate increases as a function of working memory load during the estimation/reporting phase in delayed estimation tasks although Rac-Lubashevsky et al. (2017) and Bochove et al. (2012) showed that the blink rate increases as a function of cognitive load during task switching and in incongruent trials of a flanker task, respectively. That’s the reason why we did not cite any references and instead used the expression “intriguingly” when describing the results. However, we do not agree with the reviewer’s assertion that the absence of previous report threatens the validity of our findings because it is simply the observed fact that the blink rate increased as a function of working memory load in our task, and this fact stands alone as an implication that BPR can potentially confound the main variable of interest (working memory load).

[New5]: “353-379: Something is a bit unclear in Fig 6A. It seems that the pupil size in the trials with BPR and without corrections was larger than in trials when no corrections were required. I expected to see a huge dip (due to the BPR) that would be fixed after the presented correction. However, it seems that during these trials, the pupils had no decrements that associated with BPR. If I understand correctly, this figure suggests that only one peak was found during these trials. It seems to me that something went wrong with the analysis of those trials. In addition, I'm wondering why this analysis wasn’t done on the third experiment also? In my opinion, the same test is supposed to be done on all the three experiments.”

Our reply to [New5]: To understand why the blink-affected time course of pupil size increases, rather than decreases, in Fig 6A, one needs to know the accurate definition of the “blink-affected trials.” The sentence in the fourth paragraph of the Results section titled “Correcting the auditory oddball data for BPR” of the submitted manuscript clearly defines what the “blink-free trials” and “blink-affected trials” are: “Considering that BPR lasts for about 3 s, the blink-free trials were defined as the trials whereby blinks occurred neither in the 2-back trial, in the 1-back trial, nor in the current trial while the blink-affected trials were all the rest of the trials (see Methods and Materials for details)”. That is, the blink-affected trials include a set of trials in which blinks occurred either in the 2-back trial, in the 1-back trial, or in the current trial. In the trials in which blinks occurred in the current trial, the effect of BPR appears as a dip. The effect usually affects only the second half part (1-2 s) of the trial (see 0-2 sec of the bottom panel of Fig 3B) since blink rate is concentrated at 0.5-1 s and (as described in the top panel of Fig 3B) and BPR has some delay (0.3 s) before a substantial decrease. In the trials in which blinks occurred in the 1-back trial, the BPR from 1-back trial typically reaches its negative peak at the onset of the current trial, so the baseline is spuriously decreased. As the pupil diameter at stimulus onset is set as baseline, BPR in the recovery (dilation) phase appears as a spurious increase of the first half part of the trial (see 2-4 sec of the bottom panel of Fig 3B). In the trials in which blinks occurred in the 2-back trial, the BPR affects similarly as the 1-back trial case, but with much smaller effect since BPR from the 2-back trial already substantially recovered prior to the current trial. The combination of the decrease in the last half part (from the trials in which blinks occurred in the current trial) and the increase in the first half part (from the trials in which blinks occurred in the 1-back and 2-back trial) eventually made spuriously exaggerated dilation of the baselined time course in this task. We anticipated these spurious patterns via simulation based on our generative model (Fig 5A-B), and the observed patterns supported (Fig 5C) confirmed this anticipation. The thick black line in Fig 6A, which is the same as the thick black line in the left panel of Fig 5C, is the observed time course of the “blink-affected trials”, which is quite similar to that predicted by the model simulation (the right panel of Fig 5B) and the mean of the spurious patterns (dashed and dash-dotted lines in the left panel of Fig 5C). Although we think the currently submitted manuscript provides an account for why and how the time course of the “blink-affected trials” increase (not decrease), we further elaborated this account in detail to help readers readily follow our account in the revised manuscript (the fourth paragraph of the Results section titled “Correcting the auditory oddball data for BPR”).

 As for the comment regarding the issue of “why this analysis wasn’t done on the third experiment also?”, please read our reply to [The issues related to the major point [C2-1] in the first-round of review/revision] in the above. 

[New6]: “454-466: There were differences between the conditions at time 0. Therefore, it is difficult to conclude that the observed differences after the manipulation (i.e., the presentation of the stimulus) were caused by the differences themselves. In my opinion, the authors should present relative changes (compared to the stimulus onset, for example?).

Moreover, at the trial offset (the beginning of trial n+1) the pupil size was not equal to that at the trial onset (the beginning of trial n). Due to the fact that trial n and trial n+1 are identical in terms of the presentation of the mean response, it is problematic that the trial onset wasn’t identical to the trial offset. It means that there was an influence of trial n on trial n+1 (in addition to the theoretical influence of the BPR). In other words, there was a methodological issue in the experiment. A study that deals with (relatively minor?) artifacts like BPR cannot use an experiment with a critical problem like that.”

Our reply to [New6]: By the way, we clarify that, as we explicitly wrote in the figure caption for Fig 9A, the ‘time 0’ demarcates not the stimulus onset but the onset of the estimation epoch. To address the reviewer’s point, we simply expanded the time window of plotting around from the time point when the sensory stimulus was masked, because we were interested in knowing how the peak amplitude in pupil size during the mnemonic period (i.e., the period from stimulus offset and estimation response, during which mnemonic representation is formed, maintained, and used for estimation) differ across the different memory load conditions. In this way, readers can see the developing time course of pupil size corresponding to the (mnemonic) period relevant to our purpose. In the revised manuscript, we modified the figure caption of Fig 9 and the main text (the second paragraph in the Results section titled “Correcting the delayed orientation estimation data for BPR”). 

 As for the mismatch between the trial offset and onset in pupil size in Fig 8, we clarify that the data at the end of the plot is not the data at the trial offset. As we described in Methods (lines 744-745) in the previous manuscript, the estimation epoch and the confidence report epoch varied in length from trial to trial depending on how quickly subjects estimated and judged their confidence (mean±SD and range for estimation duration, 2.11 ± 0.51 sec [min = 1.55, max = 3.85 sec]; mean±SD and range for confidence judgment duration, 0.77 ± 0.29 sec [min = 0.41, max = 1.54 sec]), resulting in varying trial duration across trials. This explains the superficial mismatch between the leftmost points and the rightmost points in pupil size in Fig 8. In the revised manuscript, we further specified the temporal structure of the delayed orientation estimation task (in Methods & Materials subsection titled “delayed orientation estimation task”).

[New7]: “504-522: The summary talks about the algorithm itself that wasn’t described at all in the manuscript. What is the purpose of the study? To describe the BPR issue? To address the BPR issue? Up until this part in the manuscript, no details about the algorithm were provided. How are the readers supposed to judge the algorithm?”

Our reply to [New7]: Because the details of the BPR correction algorithm were too lengthy and complex to be described in Results, we elected to give only a conceptual and intuitive description of the algorithm in the Results section titled “Model-based correction of pupillary responses for BPR” – just before testing its validity and effectiveness – and provide the details and corresponding mathematical equations comprising the algorithm in the Methods section titled “The algorithm of correcting pupillary responses for BPR”. We still believe that this option works best. Note that, in the submitted manuscript, we explicitly directed readers’ attention to this Methods & Materials section for the specifics and details of the algorithm (“By incorporating what we learned about BPR from the fixation task into the generative model as prior knowledge, we developed an algorithm inferring subject-specific BPR profile and blink-by-blink BPR amplitude by combining that prior knowledge and the likelihood function acquired from the observed pupil measurements (see Materials and Methods for the detailed description of the algorithm)”).

 We believe that it is more appropriate and effective to state the purpose of the study in the Introduction, not in the first paragraph of the Discussion, which only recapitulates the gist of important findings in the study before delving into an in-depth discussion of those findings. Note that the motivation, purpose, and anticipatory summary of our study are compactly and explicitly provided in the last paragraph of Introduction.

[New8]: “610-618: The authors said that the other approach is not good enough because of the two mentioned crucial aspects. However, no comparison between the approaches was applied. If the authors suggest that their approach (that wasn’t discussed in the manuscript except for in the last section) is better, a comparison should be done. In my opinion, a comparison between the 2 approaches and between the data after the BPR corrections to trials with no BPR in the three experiments has to be done.”

Our reply to [New8]: We do not agree with the reviewer’s accusation: we never stated or indicated that “the other approach is not good enough”. Here we simply indicated in what aspects the other approach (Knapen et al., 2016) and our approach differ (“However, this approach differs from ours in two crucial aspects. First, … Secondly, … In addition,…”). We cannot claim that our approach is better than that approach in the current work because it’s difficult for us to evaluate Knapen et al. (2016)’s method because, as described in the text, it requires us to build an event matrix unlike ours and outcomes would be highly affected depending on the event matrix. For this reason, we opted to let readers know how our approach differs from Knapen et al. (2016)’s method, but not to claim the superiority of our approach over it.

[New9]: “625-627: This is a serious problem. It means that the algorithm might destroy valid data with a wrong interpolation. Isn't it safer to ignore the BPR?”

Our reply to [New9]: Although we exercised scientific rigor here by bringing up a theoretical possibility that the pupil time course of a certain cognitive event matches BRP exactly in timing and shape, such cases must be very rare. Our demonstration of increases in statistical power in the auditory oddball and delayed estimation task support this. In the revised manuscript, we added a sentence to indicate that such cases are very rare.

[New10]: “641-646: In my opinion, the “back to baseline” is a good reason for a relatively long ITI. As it is reflected in Exp. 2, this might also be problematic if there are no blinks (and BPR) at all.”

Our reply to [New10]: We agree with the reviewer. We incorporated the reviewer’s point by modifying this paragraph.

[New11]: “647-652: Please note that this is also really relevant in case of blink corrections. Actually, in general it is highly recommended to measure pupil size before the manipulation onset. As reflected from Fig. 9, there are differences in the manipulation onset. Actually, we don’t really know what was happening before the manipulation onset in terms of the pupil size. Therefore, it is difficult to conclude that the manipulation itself caused the results we see. Specifically, in addition to the comparison relative to the stimulus onset, the researcher has to be sure that any changes in the pupil size didn’t start before the manipulation itself. Hence, the recommendation about recording the data before the manipulation onset is not only relevant for BPR, but for pupillometry studies in general (also for studies when there are no eye blink at all by using special drags ...).”

Our reply to [New11]: We agree with the reviewer that it’s “highly recommended to measure pupil size before the manipulation onset.” Accordingly, we incorporated the reviewer’s point by adding a sentence in this paragraph. (However, See our reply to [New5] in the above regarding the reviewer’s extended comment on Fig. 9). 

[New12]: “653-663: In contrast to RT studies (but not only), in pupillometry studies it is important to get data from the eyes. Hence, voluntary eye blinks (for a rest?) are not highly recommended. Honestly, I don’t believe that it is possible to avoid involuntary eye blinks (that are associated with all the aspects that the authors mentioned). Hence, I think this suggestion (avoid researchers to ask participants to avoid eye blinks) is wrong and might cause loss of data.”

Our reply to [New12]: We are a bit sympathetic to the reviewer’s comment: because subjects cannot completely suppress blinks even if experimenters ask them to suppress blinks, experimenters can maximize the blink-free data by instructing subjects to suppress eye blinking. However, we do not agree with the reviewer’s assertion that “this suggestion (avoid researchers to ask participants to avoid eye blinks) is wrong and might cause loss of data.” ‘An attempt of suppressing blinks intentionally’ itself, even if that’s not completely successful, comes with many cons as we listed in the manuscript: (1) loss and distortion of valuable blink data, (2) fatigue associated with blink suppression, and (3) creation of unwanted cognitive or brain states due to blink suppression. Therefore, we think it is better for experimenters not to say anything to subjects about blinks at all and preprocess BPR, as we showed in the current work. However, we admit that we have not compared the two conditions (instruct to blink naturally vs. suppressing blink), so we modified the expression from “do not recommend” to “recommend to consider not giving”. Note that the Discussion section titled “practical considerations for avoiding BPR-related problems” is a set of practical guides we can offer with the premise that our findings presented in the current work are valid. On that premise, we think it is reasonable to recommend that experimenters consider not giving the instruction of blink suppression.

[New13]: “701-702: The authors decided to not give the subjects any instructions about eye blinking. They also recommended other researchers do the same (as reflected from lines 653-663). Hence, it seems that there is a conflict of interest in their recommendation, specifically if they didn’t examine it directly; namely, if they did not compare between two groups that are different based on the instructions.”

Our reply to [New13]: We do not agree with the reviewer’s assertion that “there is a conflict of interest in their recommendation”. We did not give instruction about blinking for several reasons. Instructing subjects to suppress blinks is an act intervening in the distribution of spontaneous blinks. We intended to observe how pupil data is confounded by BPR from spontaneous blinks without intervention, so we did not give any instruction about blinking. Also, many pupillometry experiments are being conducted without instruction about blinking as well, so it shows well how pupillometry is confounded by BPR under natural and conventional circumstances. Additionally, there is a logical reason why we did not recommend instruction of blink suppression. Suppressing blinks can reduce BPR but induce other problems as we described. Therefore, we believe that it is better to let subjects blink naturally and preprocess BPR as much as possible, rather than instructing suppressing subjects’ blinking. To sum up, we have a reasonable ground about not giving instruction about blinking, and recommend other researchers to consider doing that, and we do not see any inconsistency or conflict of interest here.

[New14]: “763-766: Did the authors find meaningful differences between the two eyes? If the authors decided to use binocular measurement, why wasn’t the mean pupil size (of the two eyes) used for the analysis?”

Our reply to [New14]: Yes, there were differences in quality occasionally between the two eyes, and we opted to use the data from one of the two eyes that provided more reliable data for the following reasons. First, there were subtle but noticeable differences in blink offset timing between the two eyes. Because we wanted to define the time course of BPR precisely, we were concerned that averaging two signals that differ in timing might introduce unwanted blurs. Second, occasionally in some individuals, only one eye’s data was not detected or became unstable probably due to eyelid occlusion, which is known to occur more frequently in Asian people. If we average the signals from the two eyes, the time courses are expected to show weak but spurious fluctuations due to the missed signals from the unreliable eye. In the revised manuscript, we stated the reason why we opted to analyze one eye’s signal for pupil data (the first paragraph in the section titled “Data analysis”). 

[New15]: “785-787: In my previous review I asked why the smoothing was required. I understand that the authors asked to reduce the noise of the data, however, smoothing with a boxcar time window of 250 ms not only reduces the noise, but also the signal itself. Actually, a boxcar time window of 10 ms should remove most of the noise. Therefore, it seems to me that the smoothing was too aggressive and in my opinion should be a new concern. EyeLink devices are well known to be high quality devices. Hence, smoothing is not really required in pupillometry studies. For reference, in our lab we are using the eye tribe devices (that have a sampling rate of 60 Hz and are well known to be relatively noisy devices). After blink correction, no smoothing is required. The reason that I mention it again (after I asked about it in my previous review) is due to the data in Figure 6. In this figure, the data looks too processed and it made me question whether the smoothing might have caused these patterns.” (related to [G4])

[New16]: “794-795: If the reason behind the smoothing is to remove noise, why was the band-pass filter also required? Was the smoothing used only for the detection of the blinks? It is not clear to me.”

Our reply to [New15] & [New16]: We clarify that the smoothing with a boxcar time window (0.25 sec) was applied only to define the endpoint of the post-blink artifact (for the purpose of being not too sensitive for detection), but was never applied to the pupil time course that was used for analysis. The data used for analysis were smoothed only by the band-pass filter (0.02Hz and 4Hz with the 3rd order Butterworth filter), which is often practiced in other pupillometry studies. To avoid confusion, we added a sentence to stress that the boxcar smoothing was not applied to the data used for analysis (section titled “The removal of artifacts around single blink events”). 

 As for the impression that “the data [in Fig 6.] looks too processed”, we think it looks smooth because, firstly, the lines represent the averages across many subjects (n=27) and many trials/subject (n=980~1,260 trials) and, secondly, (as we just mentioned above), the high-frequency noises were filtered out with the bandpass filter.

[New17]: “817-826: Similar to the general comment about the separation between trials, I don’t really understand this rationale. In each trial there were two options—there was a correction of BPR (by using the authors’ algorithm) or there was no correction. Why do the authors separate the trials differently?”

Our reply to [New17]: This is the same comment as [C2-1]. Please see above for our reply on this issue (Our re-reply to the reviewer’s rebuttal to our reply to [C2-1] in the first round of revision: Regarding the issue of “applying the same analysis to the three experiments”.

---

## [Editor Report · Decision Letter 2]

3 Dec 2021

The Confounding Effects of Eye Blinking on Pupillometry, and their Remedy

PONE-D-21-12636R2

Dear Dr. Lee,

We’re pleased to inform you that your manuscript has been judged scientifically suitable for publication and will be formally accepted for publication once it meets all outstanding technical requirements.

Kind regards,

Manuel Spitschan

Academic Editor

PLOS ONE
---

## [Editor Report · Acceptance letter]

9 Dec 2021

PONE-D-21-12636R2 

The Confounding Effects of Eye Blinking on Pupillometry, and their Remedy 

Dear Dr. Lee:

I'm pleased to inform you that your manuscript has been deemed suitable for publication in PLOS ONE. Congratulations! Your manuscript is now with our production department. 

Kind regards, 

on behalf of

Dr. Manuel Spitschan 

Academic Editor

PLOS ONE